

# On the self-regulating effect of grain size evolution in mantle convection models: Application to thermo-chemical piles

Jana Schierjott[1], Antoine Rozel[1], and Paul Tackley[1]

[1]Institute for Geophysics, Department of Earth Sciences, Sonneggstrasse 5, 8092 Zurich, Switzerland

**Correspondence:** Jana Schierjott (jana.schierjott@erdw.ethz.ch)

**Abstract.** Seismic studies show two antipodal regions of lower shear velocity at the core-mantle boundary (CMB) called Large Low Shear Velocity Provinces (LLSVPs). They are thought to be thermally and chemically distinct, and therefore might have a different density and viscosity than the ambient mantle. Employing a composite rheology, using both diffusion and dislocation creep, we investigate the influence of grain size evolution on the dynamics of thermo-chemical piles in evolutionary
geodynamic models.

We consider a primordial layer and a time-dependent basalt production at the surface to dynamically form the present-day chemical heterogeneities, similar to earlier studies, e.g., by Nakagawa and Tackley (2014). We perform a parameter study which includes different densities and viscosities of the imposed primordial layer. Further, we test the influence of yield stress and parameters of the grain size evolution equation on the dynamics of piles and their interaction with the ambient mantle.

Our results show that, relative to the ambient mantle, grain size is higher inside the piles, but due to the large temperature at the CMB, the viscosity is not remarkably different from ambient mantle viscosity. We further find, that although the average viscosity of the detected piles is buffered by both grain size and temperature, grain size dominates the viscosity development. However, depending on the convection regime, in the ambient mantle, viscosity can be dominated by temperature.

All pile properties, except for temperature, show a self-regulating behaviour: although grain size, density and viscosity
decrease when downwellings or overturns occur, these properties quickly recover and return to values prior to the downwelling. We compute the necessary recovery time and find, that it takes approximately 400 Myr for the properties to recover after a resurfacing event. Extrapolating to Earth-values, we estimate a much smaller recovery time.

We observe that dynamic recrystallisation counteracts grain growth in the piles when the lithosphere is weakened and forms downwellings. Venus-type resurfacing episodes reduce the grain size in piles and ambient mantle to few millimetres. More
continuous mobile-lid type downwellings limit the grain size to a centimetre. Consequently, we find that grain size-dependent viscosity does not increase the resistance of thermo-chemical piles to downgoing slabs. Mostly, piles deform in grain size-sensitive diffusion creep but they are not stiff enough to counteract the force of downwellings. Hence, we conclude that the location of subduction zones could be responsible for the location and stability of the thermo-chemical piles of the Earth because of dynamic recrystallisation.





## 1 Introduction

Seismic studies show two antipodal regions of low shear velocity at the core-mantle boundary (CMB), one beneath the Pacific and one beneath parts of Africa and the Atlantic (Ritsema et al., 2011; Lekic et al., 2012; Garnero et al., 2016). These regions, called Large Low Shear Velocity Provinces (LLSVPs), are thought to be thermally and chemically distinct and thus, differ in density and viscosity from the surrounding material (Masters et al., 2000; Ishii and Tromp, 1999; Trampert et al., 2004).

The shape of LLSVPs is relatively well constrained thanks to seismic tomography models. They consistently reveal a roundish shape for the African LLSVP and an overall north-south elongated form for the Pacific LLSVP (Ritsema et al., 1999; Kuo et al., 2000). In total LLSVPs cover around 20 - 50 % of the area at the CMB (Burke et al., 2008; Garnero and McNamara, 2008) and make up between roughly 1.6 - 2.4 % of the total mantle volume (Burke et al., 2008; Hernlund and Houser, 2008). The African LLSVP extends upward from the CMB about 1000 km; the height of the Pacific one is less well constrained but is in any case smaller with about 400-500 km of upward extension (Garnero and McNamara, 2008). Following Torsvik et al. (2006, 2010) LLSVPs have not changed their position for at least 200 Myr, possibly up to 540 Myr.

Apart from the geometry other properties of LLSVPs are not that well defined. The negative correlation between bulk sound speed and shear wave velocity suggests a chemical origin (Masters et al., 2000; Trampert et al., 2004) of LLSVPs. Normal-mode data support a density increase of a few percent compared to the ambient mantle (Ishii and Tromp, 1999; Trampert et al., 2004). Recently though, Koelemeijer et al. (2016) proposed that LLSVPs might rather exhibit a reduced density. Using deep mantle-sensitive Stoneley mode data in a joint P- and S-wave inversion this recent work showed that LLSVPs might be lighter than the ambient mantle, but only above 100 km upward from the CMB. Chemical heterogeinities and/or the presence of post-perovskite (pPv) and its interplay with the thermal boundary layer could explain the observations.

Laboratory studies, e.g. by Davaille et al. (2005) are able to mimic the 3D-complexity of LLSVPs and, as numerical models, provide insight into the development over time. Seismological studies on the other hand, can only provide information on LLSVPs for the current time snap. Davaille et al. (2005) emphasized in their work that the presently observed upwellings are all of transient nature and that all types such as plumes, LLSVPs and traps might represent different stages of the same evolving thermo-chemical instability. Nevertheless, they also suggest that the upwellings are of different chemical composition.

Also concerning their origin, researchers have suggested various hyptheses. LLSVPs might originate from recycled sub-ducted slabs, from survived remenants of reservoirs from the early partial differentiation of the mantle (Deschamps et al., 2015) or from a mix of both (e.g. basal mélange (BAM), (Tackley, 2012)). Recently, Ballmer et al. (2016) suggested that even LLSVPs themselves could consist of two different types of materials, an upper basaltic and a lower primordial one. Since LLSVPs remain physically unreachable numerical and experimental studies try to constrain the parameter space. McNamara and Zhong (2004) numerically studied how the top surface of LLSVPs changes with different compositions and were able to show that LLSVPs consisting of accumulated crust have a rougher or more diffusive top than the ones made up of primordial material.

In numerical studies both, a lower e.g., (McNamara and Zhong, 2005) and a higher viscosity e.g., (McNamara and Zhong, 2004) have been investigated. We learn from McNamara and Zhong (2004) that the viscosity contrast between different com-





ponents could well be the main control on how the piles in the lowermost mantle are organized. In their study they find that an
intrinsic viscosity increase of dense material in the bottom of the mantle yields fewer but larger piles than only a temperature-
dependent rheology. However, most of the works on thermo-chemical piles have in common, that viscosity is treated either
depth- or/and temperature-dependent. Only very few studies have considered a composite and grain size-dependent viscosity
(Hall and Parmentier, 2003; Solomatov and Reese, 2008; Dannberg et al., 2017).

Not only numerical modelers come to the conclusion that grain size-dependent viscosity should be used in future studies
(Yang and Fu, 2014). Rather, the idea originates from experimentalists who have shown how important it might be to include
grain size evolution in the viscosity formulation (Karato and Wu, 1993; Karato, 2010). In experiments they observe grain size
reduction under high strain, (e.g. Karato et al., 1993) and grain growth when conditions favor high grain boundary energy
(Karato, 1989). In times of high stress dynamic recrystallization operates leading to a smaller grain size and shifting the
deformation regime from dislocation to diffusion creep. As a result, regions under the influence of high stress exhibit a lower
viscosity than the surrounding regions (Warren and Hirth, 2006).

Karato et al. (1995) suggest that most parts of the lower mantle likely deform under diffusion creep due to the absence of
shear wave splitting. However, several other studies indicate that in many regions dislocation creep is active (Lay et al., 1998;
McNamara et al., 2001). Among others, Cordier et al. (2004) suggested dislocation creep as the deforming mechanism for the
perovskite phase in the uppermost lower mantle and McNamara et al. (2001) for regions around downwellings. Therefore, it
would be worth to not only consider grain size-dependent diffusion creep but additionally a composite rheology formulation
involving both diffusion and dislocation creep. Since dislocation creep is favoured when grain sizes are large, the region along
the CMB, hot upwellings and plumes might rather deform in dislocation creep because temperature and stresses are high
(Solomatov and Moresi, 1996; Karato and Rubie, 1997; Solomatov et al., 2002; Korenaga, 2005).

The wide range of proposed possibilities in terms of composition, viscosity and density of LLSVPs convinced us to apply
the grain size-dependent, composite viscosity formulation implemented in the global convection code StagYY for studying the
effects on the development of LLSVPs. By also considering a primordial layer we are able to elaborate on the origin of LLSVPs
(e.g. subducted basalt, primordial reservoir or the basal mélange (Tackley, 2012)), and the interaction between subducted basalt
and the primordial layer. Moreover, we study how thermo-chemical piles behave in the dynamic system of mantle convection
using simulations evolving over 4.5 billion years. We investigate whether piles behave as obstacles to convection, whether they
get pushed around or even entrained by mantle flow.

## 2 Model

### 2.1 Setup

Apart from the rheology, our model set up is very similar to the model used by Nakagawa and Tackley (2014). The composition
of the mantle consists of 80% harzburgite and 20% basalt. In other words, the pyrolitic composition is a mechanical mixture
of 60% olivine and 40% pyroxene-garnet phases. Phase transition depths, temperatures, densities and Clapeyron slopes for





the independent olivine and pyroxene-garnet phases can be found in table 1. Additionally, we impose a primordial layer with physical properties similar to pyroxene-garnet at the base of the mantle. The initial temperature at the CMB is set to 5000 K, at the surface to 300 K.

Further, melting and crustal production in the simplified two-phase system is included. Melting helps buffering the internal temperature of the Earth (Armann and Tackley, 2012) and affects the tectonic regime as it generates compositional heterogeneities (Lourenço et al., 2016, 2018). Typically, melting of the pyrolitic mantle locally produces molten basalt and a solid residue more enriched in harzburgite than the source rock. If the melt is generated at a depth lower or equal to 300km, the basalt is either erupted at the surface of the model or intruded at the base of the crust. When the melt is erupted, it is assumed it

cools instantly to surface temperature. When the melt is intruded, only adiabatic cooling is subtracted from it while it is brought upward. Intruda is therefore warmer than the ambient lithosphere which results in lithosphere-weakening. We use a constant partitioning of eruption as opposed to intrusion. The fraction of eruption is called 'eruption efficiency' ($er$) and has been shown to have a strong influence on the thermal states of both mantle and lithosphere (Lourenço et al., 2018). In conjunction with testing the eruption efficiency, we test more parameters which influence the convection regime such as the yield stress ($\tau_y$) and

the yield stress gradient ($c_{\tau_y}$).

To account for the compressibility of mantle material, we use a third order Birch-Murnaghan equation of state. A detailed explanation and list of parameters can be found in Tackley et al. (2013). All solid phases have a bulk modulus of 210 GPa in the lower mantle, 85 GPa in the transition zone, and a bulk modulus of 163 GPa in regions shallower than the transition zone. Solid phases also have a bulk modulus gradient which is 3.9 in the lower mantle and 4 everywhere else. A Grüneisen parameter

of 0.85 is used in the transition zone and 1.3 everywhere else. Molten phases (molten basalt and molten harzburgite) have everywhere a bulk modulus of 30 GPa, a bulk modulus gradient of 6 and a Grüneisen parameter of 0.6. The surface densities of each phase are given in table 1.

To study the evolution of Large Low Shear Velocity Provinces (LLSVPs) we impose a 200 km thick basal primordial layer along the CMB in the beginning of the runs. The physical properties of the primordial layer are the same as basalt but with a

different viscosity (see equation 6) and density (table 1 & 2). In order to test the dynamic effect of the density of primordial material, we vary its surface value. When $\rho_{\mathrm{prim}} = 3080$ kg/m$^3$, the primordial material has the same density as the basalt phase. When $\rho_{\mathrm{prim}} = 3140$ kg/m$^3$, the primordial material is 60 kg/m$^3$ denser than the basalt phase, all the way between the surface and the CMB. In other words, the difference of the primordial layer's density and the ambient mantle's density is defined by the buoyancy number (Le Bars and Davaille, 2002)

$$B_{\mathrm{prim}} = \frac{\rho_1 - \rho_2}{\rho_0 \alpha \Delta T} \tag{1}$$

where $\rho_0 = \frac{\rho_1 + \rho_2}{2}$, with $\Delta T$=4700 K as the average temperature difference between the bottom and the top of the model-domain at the beginning of the run time, $\alpha$ as the thermal expansivity, $\rho_1$ as the density of primordial material and $\rho_2$ as the density of basalt. $\rho_1$ and $\rho_2$ can be found in table 2 as the surface density of primordial material and basalt, respectively.





In addition to pile-related parameters, we vary the intensity of dynamic recrystallisation (see term $f_{\text{top}}$ in equation 14), and
the diffusion creep efficiency in the upper and lower mantle ($\chi_{UM}$ & $\chi_{LM}$) to investigate their effect on mantle convection in
general (table 2). A compilation of all models can be found in table 5. The bold-marked models are used for specific figures
in the result section. We emphasize that the used simulations either represent average observations, or show the extreme.
Generally, the result section shows that the effective quantities such as viscosity, grain size, rheology and stress in the deep
mantle weakly depend on the input parameters. This can be understood by the interesting presence of self-regulating processes
also discussed in the result section.

## 2.2 Conservation of mass, momentum and energy

We use a thermo-mechanical modelling approach in 2D-spherical annulus geometry (Hernlund and Tackley, 2008) to model the
development and evolution of thermo-chemical piles along the CMB. We solve the conservation equations for a compressible
fluid using a finite difference method on a fully staggered grid (Tackley, 2008; Hernlund and Tackley, 2008). Pressure, density
and viscosity are defined in the cell-centres whereas velocities are placed on the cell edges. Temperature, composition, grain
size and additional material attributes are tracked using Lagrangian tracers which are moved according to the velocity field and
extrapolated to the cell centres. The computational domain consists of $512{\times}64$ cells, with a radially varying resolution which
is higher at the surface, the 660 km phase transition, and along the CMB.

In the anelastic approximation, density, expansivity, diffusivity and heat capacity are functions of depth, and the Prandtl
number is considered infinite (Tackley, 2008). Mass conservation is written as

$$\div(\boldsymbol{v}\rho) = 0 \tag{2}$$

with velocity $\boldsymbol{v}$ and density $\rho$.

The equation for conservation of momentum is

$$\boldsymbol{\nabla}{\cdot}\boldsymbol{\tau} - \boldsymbol{\nabla}P = -\rho(C,r,T)\boldsymbol{g} \tag{3}$$

where $\tau$ is the deviatoric stress tensor, $P$ is pressure, density depends on composition $C$, temperature $T$ and radius $r$, and $\boldsymbol{g}$ is
the gravitational acceleration.

Conservation of energy is defined as

$$\rho C_{\text{p}}\left(\frac{\partial T}{\partial t} + \boldsymbol{v}\cdot\boldsymbol{\nabla}T\right) = \alpha T(\boldsymbol{v}_r\cdot\boldsymbol{\nabla}_r P) + \boldsymbol{\nabla}{\cdot}(\kappa\boldsymbol{\nabla}T) + \rho H + \boldsymbol{\tau}{:}\boldsymbol{\nabla}\boldsymbol{v} \tag{4}$$

with radial velocity $\boldsymbol{v}_r$, internal heating rate per unit mass $H$, specific heat capacity $C_{\text{p}}$, and $\kappa$ as the thermal conductivity. The
first term on the right-hand side is the heat production/consumption due to adiabatic (de)compression, the second describes
heat diffusion, the third term contributes radiogenic heating and the fourth term adds viscous dissipation during non-elastic





deformation processes (Ismail-Zadeh and Tackley, 2010). The viscosity $\eta$ varies with temperature, depth, strain rate or stress, composition and grain size. For details on our viscosity formulation see the following sections.

## 2.3 Rheology

We use a visco-plastic modelling approach. The viscous deformation can be accommodated by two mechanisms: diffusion and dislocation creep. Diffusion creep is grain size-sensitive and diffusion creep strain rate is directly proportional to shear stress. Dislocation creep is a non-Newtonian deformation mechanism where strain rate and applied shear stress are related via power law. Both creep mechanisms depend on temperature (activation energy) and pressure (activation volume) of the system (Ranalli, 1995). The total strain rate $\dot{\epsilon}_{tot}$ is a sum of the strain rate in dislocation $\dot{\epsilon}_{ds}$ and diffusion creep $\dot{\epsilon}_{df}$ (Weertman, 1970;

Frost and Ashby, 1982; Hall and Parmentier, 2003). Following the fundamental relation between stress and strain rate tensors $\tau = 2\eta\dot{\epsilon}$, we can identify the dislocation and diffusion creep components of the viscosity:

$$\eta_{\text{ds}} = \frac{\Delta\eta_{\text{ds}}\eta_{\text{prim}}}{2A_{\text{ds}}} \exp\left(\frac{E_{\text{ds}} + PV_{\text{ds}}}{RT}\right)\tau^{1-n} \tag{5}$$

$$\eta_{\text{df}} = \frac{\Delta\eta_{\text{df}}\eta_{\text{prim}}}{2A_{\text{df}}} \exp\left(\frac{E_{\text{df}} + PV_{\text{df}}}{RT}\right)\mathcal{R}^{m}, \tag{6}$$

where $\Delta\eta_i$ are dimensionless constants used to impose viscosity jumps at the 660-discontinuity for each creep mechanism.

$\Delta\eta_i$ are equal to 1 in the upper mantle and are greater than 1 in the lower mantle. $\eta_{\text{prim}}$ is only different from 1 in the primordial material. $A_i$ are rheological prefactors, $E_i$ and $V_i$ are activation energies and volumes, respectively. $\mathcal{R}$ is the average grain size (see equation 12), $\tau$ is the second invariant of the shear stress, $n$ is the dislocation creep exponent, $m$ is the diffusion creep grain size exponent. Rheological coefficients depend on the creep regime but not on composition (see Table 1).

In order to study the importance of the relative contributions of diffusion and dislocation creep, we define the composite

viscosity using their weighted contributions:

$$\eta_{\text{creep}} \;=\; \left(\frac{\chi}{\chi+1}\frac{1}{\eta_{\text{df}}(\mathcal{R},T)} + \frac{1}{\chi+1}\frac{1}{\eta_{\text{ds}}(\boldsymbol{\tau},T)}\right)^{-1}, \tag{7}$$

where the diffusion creep efficiency $\chi$ is a dimensionless positive weight which can have a different value in the upper mantle ($\chi_{UM}$) and in the lower mantle ($\chi_{LM}$). $\chi$ greater than 1 favours diffusion creep. The equation is formulated in such a way that the value of each component of the composite viscosity (i.e., either $\eta_{\text{df}}$ or $\eta_{\text{ds}}$) corresponds to the viscosity expected for the

Earth. The sum of diffusion and dislocation creep weights is always 1, the effective viscosity is therefore not affected by the choice of $\chi$, and is usually roughly equal to the dominant viscosity. The rheological coefficients $\Delta\eta_i$, $A_i$ and $V_i$ were obtained using a semi analytical approach which ensures that the resulting effective viscosity in both diffusion and dislocation creep should be close to $10^{21}$Pa·s in the upper mantle and $10^{23}$Pa·s in the lower mantle.

The plastic rheology is employed by the use of a yield strength. The maximum strength the lithosphere can sustain is given

by a yield stress ($\tau_y$). If the yield stress is overcome, the viscosity is reduced. The yield stress consists of a brittle and a ductile





component:

$$\tau_y = \min(\tau_{\text{y,ductile}}, \tau_{\text{y,brittle}}). \tag{8}$$

The brittle yield stress follows a Byerlee law-type formulation and increases with pressure:

$$\tau_{\text{y,brittle}} = c_f P, \tag{9}$$

where $c_f$ is the friction coefficient. The ductile yield stress also linearly increases with pressure, but additionally incorporates the surface ductile yield stress $\tau_{\text{y,surf}}$ in the strength formulation, which looks similarly to the Mohr-Coulomb friction criterion:

$$\tau_{\text{y,ductile}} = c_{\tau_y} P + \tau_{\text{y,surf}}, \tag{10}$$

where $c_{\tau_y}$ is the yield stress gradient. In case the convective stresses overcome the yield stress, the viscosity is reduced to the
plastic viscosity $\eta_{\text{pl}}$, because the effective viscosity is calculated as:

$$\eta_{\text{eff}} = \min(\eta_{\text{creep}}, \eta_{\text{pl}}), \tag{11}$$

where $\eta_{\text{pl}} = \tau_y/2\dot{\epsilon}$ with $\dot{\epsilon}$ as the second invariant of the strain rate tensor.

### 2.4   Grain size evolution

In order to compute the viscosity resulting from the combined use of both creep deformation mechanisms, we perform a
number of steps. First, we calculate the grain size which we afterwards use to compute the diffusion creep viscosity. Then, we take the inverted sum of dislocation and diffusion creep viscosities to receive the total viscosity. We consider a simple grain size evolution equation in which growth and dynamic recrystallisation are competing. The experimental coefficients used (Hiraga et al., 2010) lead to a rather slow grain growth as expected in a multiphase material. The dynamic recrystallisation term has been derived in Rozel et al. (2011) and is here re-parametrised and used in a systematic way. The change of the average grain
size $\mathcal{R}$ with time is given by

$$\frac{d\mathcal{R}}{dt} = \frac{G}{p\mathcal{R}^{p-1}} - \frac{\lambda_3}{\lambda_2}\frac{\mathcal{R}^2}{3\gamma}f_G\Psi \tag{12}$$

where $\gamma$ is the surface tension, $G$ is the coarsening coefficient, $\mathcal{R}$ is the grain size, $p$ the grain coarsening exponent and $\Psi = \tau{:}\dot{\epsilon}$ the full mechanical work. $G$ is defined as follows

$$G = k_0 \exp\left(-\frac{E_G}{RT}\right) \tag{13}$$





with the universal gas constant $R$, an experimental prefactor $k_0$ and the activation energy $E_G$.

$f_G$ is the partitioning factor which determines how much of this work is used to create new grain boundaries:

$$f_G = f_{\text{top}} \left( \frac{f_{\text{bot}}}{f_{\text{top}}} \right)^{\frac{T-300}{T_{\text{CMB}}-300}} \tag{14}$$

where $T_{\text{CMB}} = 4000$ K is the average temperature at the core-mantle boundary, $f_{\text{top}}$ is the maximum (at 3000 K) and $f_{\text{bot}}$ the

minimum damage fraction (at 4000 K). In order to set the damage fraction to zero at surface temperatures of 300 K, the term

in (14) uses $-300$ in the exponent. Composition-dependence is neglected in our grain size evolution formulation, but phase

transitions are considered by resetting the grain size to 5 $\mu$m at a phase transition. All grain size evolution-related and general

model parameters are listed in table 1.

When recrystallisation and grain growth are balanced, the change of grain size with time is zero; $\frac{d\mathcal{R}}{dt} = 0$. The grain size

under this steady-state condition is referred to as equilibrium grain size $\mathcal{R}_{\text{eq}}$:

$$\frac{G}{p\mathcal{R}_{\text{eq}}^{p-1}} = \frac{\lambda_3}{\lambda_2} \frac{\mathcal{R}_{\text{eq}}^2}{3\gamma} f_G \tau : \dot{\epsilon} \tag{15}$$

$$\Leftrightarrow \mathcal{R}_{\text{eq}} = \left( \frac{3\gamma G \lambda_2}{p f_G \tau : \dot{\epsilon} \lambda_3} \right)^{\frac{1}{p+1}}. \tag{16}$$

Since, theoretically, the stress state of rocks can be reconstructed from a given grain size and known temperature, this state is

called piezometer or wattmeter (De Bresser et al., 1998; Austin and Evans, 2007; Rozel et al., 2011).

## 2.5 Primordial layer and pile detection

The pile-detection is based on composition and time-dependent temperature. At least 90% of the pile must consist of primordial

material ($C_{prim}$) and/or basalt ($C_{bas}$) :

$$C_{prim} + C_{bas} > 0.9 \tag{17}$$

The temperature constraint is defined as the average of a mid-mantle temperature of 3000 K and the current CMB-temperature:

$$T_{pile} \geq (3000K + T_{CMB}). \tag{18}$$

If one of the criteria is not fulfilled, the pile top is reached (figure 1). At each time step average values for properties such as

viscosity, density, temperature and grain size of the pile are computed. Additionally, 1D-profiles through the pile and through

the ambient mantle are calculated.

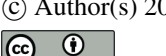



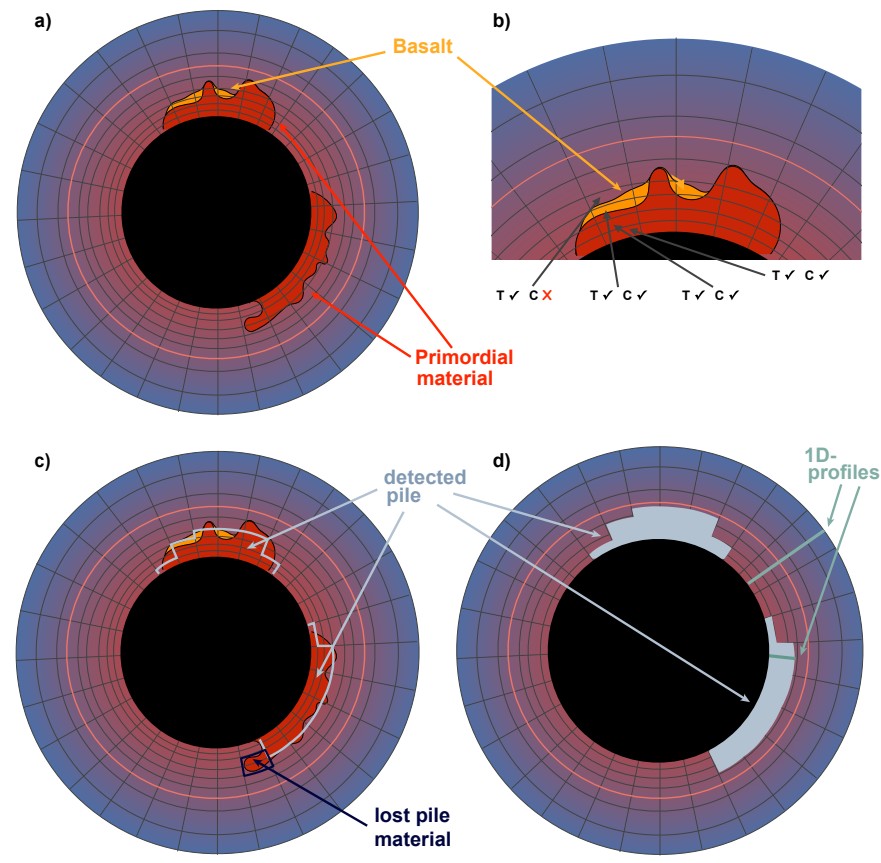

**Figure 1.** Sketch showing the steps of our pile detection routine: First, we set the criteria, then check each cell-column starting at the CMB for the criteria and stop the detection if one of the criteria is not any more fulfilled. Finally, we write a new pile-field whose characteristics are saved and can be used for further postprocessing.

## 3 Results

In the current section, we chose to first illustrate the effect of grain size evolution on the dynamics of thermo-chemical piles mainly using the various convection regimes depicted in simulation number 72. This case is of particular interest as it nicely represents the diversity of processes experienced in all the other simulations: starting in stagnant lid regime, experiencing basalt dripping stages, resurfacing episodes and a rather long mobile lid regime phase (the closest to plate tectonics behaviour of the Earth). Simulations number 3, 7 and 73 are also used to illustrate the competing impacts of grain size and temperature on the viscosity in 0D-averages and 1D-profiles.

The result section is divided into four subsections:

(1) Dynamics of piles (2D-fields)


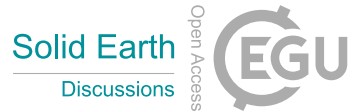

(2) Averages of pile properties over time (0D)

(3) Effect of grain size and temperature on the viscosity with focus on piles (0D)

(4) Difference between properties of pile and ambient mantle (1D-profiles)

### 3.1 The Dynamics of Piles in response to the ambient Mantle and Lithosphere

We start off by providing an overview of the dynamics of the modelled thermo-chemical piles and show results from model No 72 (table 5). In this model a yield stress of 20 MPa, a yield stress gradient of 0.1, an eruption efficiency of 0.7 and a primordial layer with a density of 3140 kg/m$^3$ ($B_{\mathrm{prim}} = 0.14$) at surface are employed. $\chi_{UM}$ and $\chi_{LM}$ are both 1, so diffusion creep and

dislocation creep are both equally important.

In figure 2, viscosity, grain size, strain rate, stress, rheology and temperature fields of time step 1.50 Gyr are shown. The rheology is defined as the ratio of strain rate due to dislocation creep and strain rate due to diffusion creep rheo $= \dot{\epsilon}_{\mathrm{ds}}/\dot{\epsilon}_{\mathrm{df}}$. If dislocation and diffusion creep equally contribute to deformation, the rheology is equal to one. Figure 3 shows snapshots of the same simulation and shows the dynamics of grain size and viscosity during an overturn event (1.58 Gyr), during the mobile

lid-phase (2.46 Gyr) and during the stagnant lid-phase (4.0 Gyr). The white line outlines the pile, the black line regions with a partial melt percentage higher than 50%. In the bottom row, the evolving distribution of basalt is presented.

Figure 2e displays the general rheology of the Earth: the lithosphere mainly deforms in diffusion creep. Small grains (around 5 $\mu$m) and a high viscosity ($10^{27}$ Pa s) mark this region. Up to 660 km, dislocation creep governs the deformation. Grains are larger (300 to 500 $\mu$m) and the viscosity is on the order of $10^{21}$ Pa s. The mid- and lower mantle is characterized by diffusion-

dominated creep. Exceptions are plumes, areas surrounding downwellings and some regions of the piles.

Downwellings lead to a very high strain rate in the surrounding material ($5\times10^{-13}$ s$^{-1}$) and consequently to a lower viscosity ($10^{20}$ Pa s) than in the ambient mantle. The grain size in the region around the downwelling is smaller (100 to 500 $\mu$m) due to the higher stress resulting in a stronger grain damage and the advection of material through phase transitions. As can be observed in figure 2, the strong, cold, basaltic material coming down from the surface has a small grain size and high viscosity. Once the

cold material reaches the lowermost mantle it destroys the pile but does not mix with it (figure 3, bottom). The downwellings force the pile to move aside and rearrange itself. The newly formed parts of the pile mainly deform in dislocation creep. The rest of the pile along the CMB mostly deforms in diffusion creep (figure 2e).

We find that piles are pushed around by downwellings but are not affected by regular convection of the ambient mantle. Piles appear to be strong as long as no force acts on them which can be attributed to the non-linearity of non-Newtonian fluids. It can

also be observed, that after a certain time, grains have grown back and reach the size they were before the overturn event. The average viscosity of the pile also returns to the previous value (figure 3). This specific time is further discussed in paragraph 3.2.4.

The subducted basaltic material accumulating along the CMB tends to melt earlier than pile- or harzburgitic material where-fore partial melt builds up where the slabs reach the CMB ($> 50$ % (black outlined region at 2.46 Gyr in figure 3). Once the

basaltic material has warmed up and mixed with the ambient mantle the pile can settle again along a larger area of the CMB.




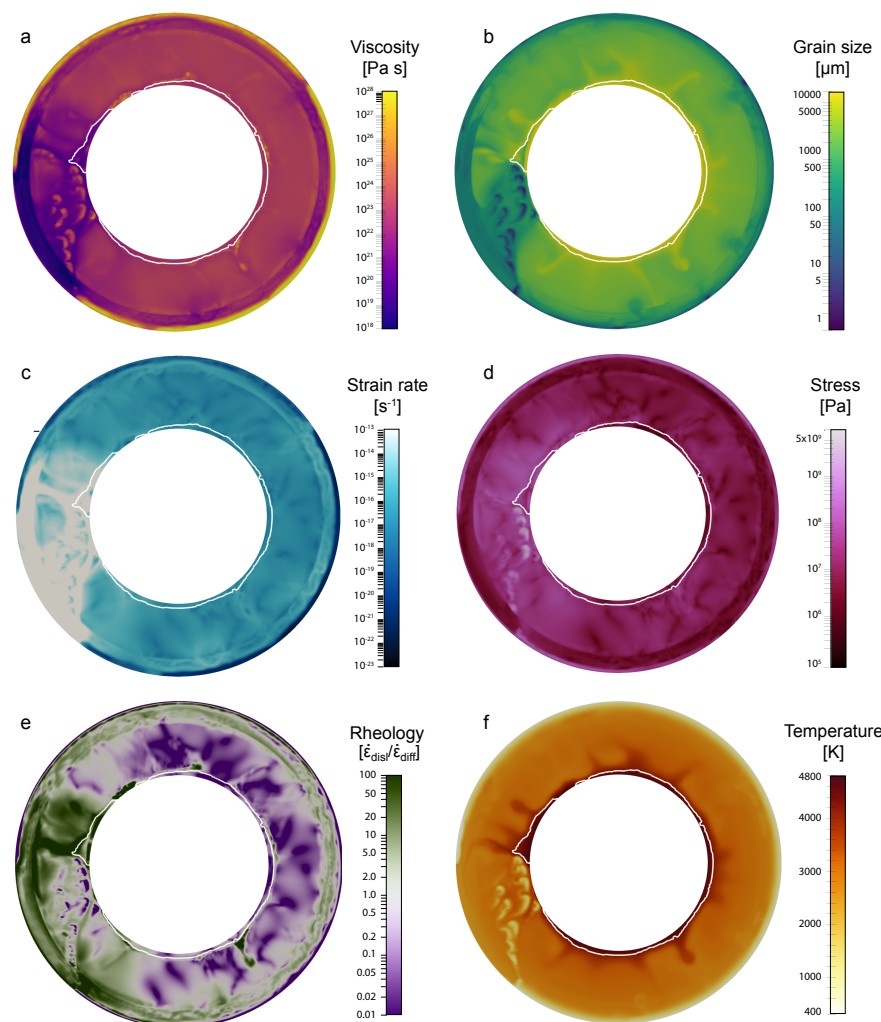

**Figure 2.** Snapshots of mantle dynamics at 1.5 Gyr. The white line confines the detected pile. A downwelling pushes the pile material around. The downgoing material is characterized by a high viscosity, very small grain size and low temperature. It mainly deforms in diffusion creep, as does most of the mantle. Only the upper mantle and parts of the pile accommodate more deformation in dislocation creep. Strain rate in the mantle surrounding the downwelling is very high and viscosity surrounding the downwelling is very low.



**Table 1.** List of grain size-related and general model set-up parameters. Grain size parameter are taken from Yamazaki et al. (2005).

| Parameter | Symbol | Value | Units |
|---|---|---|---|
| **Model parameters** | | | |
| CMB temperature (initial) | $T_{\mathrm{CMB}}$ | 5000 | K |
| Surface temperature | $T_{\mathrm{surf}}$ | 300 | K |
| Surface thermal expansivity | $\alpha$ | $3.0 \times 10^{-5}$ | 1/K |
| Phase transition depths: olivine | $d_{\mathrm{ol}}$ | 2740/660/410 | km |
| Phase transition depths: primordial | $d_{\mathrm{prim}}$ | 2740/720/400/40 | km |
| Phase transition depths: basalt | $d_{\mathrm{bs}}$ | 2740/720/400/40 | km |
| Phase transition temperature: olivine | $T_{\mathrm{ol}}$ | 2300/1900/1600 | K |
| Phase transition temperature: primordial | $T_{\mathrm{prim}}$ | 2300/1900/1600/1000 | K |
| Phase transition temperature: basalt | $T_{\mathrm{bs}}$ | 2300/1900/1600/1000 | K |
| Density changes at phase transitions: olivine | $\Delta\rho_{\mathrm{ol}}$ | 61.6/400/180 | kg/m$^3$ |
| Density changes at phase transitions: primordial | $\Delta\rho_{\mathrm{prim}}$ | 61.6/400/150/350 | kg/m$^3$ |
| Density changes at phase transitions: basalt | $\Delta\rho_{\mathrm{bs}}$ | 61.6/400/150/350 | kg/m$^3$ |
| Clapeyron slope at phase transitions: olivine | $\Gamma_{\mathrm{ol}}$ | 10/-2.5/2.5 | MPa/K |
| Clapeyron slope at phase transitions: primordial | $\Gamma_{\mathrm{prim}}$ | 10/1/1/1.5 | MPa/K |
| Clapeyron slope at phase transitions: basalt | $\Gamma_{\mathrm{bs}}$ | 10/1/1/1.5 | MPa/K |
| Friction coefficient | $c_f$ | 0.01 | |
| Surface density: solid olivine | $\rho_{\mathrm{s,ol}}$ | 3240 | kg/m$^3$ |
| Surface density: solid pyroxene-garnet | $\rho_{\mathrm{s,pg}}$ | 3080 | kg/m$^3$ |
| Surface density: molten olivine | $\rho_{\mathrm{m,ol}}$ | 2900 | kg/m$^3$ |
| Surface density: molten pyroxene-garnet | $\rho_{\mathrm{m,pg}}$ | 2900 | kg/m$^3$ |
| | | | |
| **Diffusion and dislocation creep parameters** | | | |
| Activation volume | $V_{df}$ | $5.5 \times 10^{-7}$ | m$^3$/mol |
| Activation energy | $E_{df}$ | $3.75 \times 10^5$ | J/mol |
| Prefactor | $A_{df}$ | see table 5 | |
| Viscosity jump | $\Delta\eta_{df}$ | see table 5 | |
| Grain size exponent diffusion creep | $m$ | 3.0 | |
| Activation volume | $V_{ds}$ | $2.9 \times 10^{-7}$ | m$^3$/mol |
| Activation energy | $E_{ds}$ | $5.3 \times 10^5$ | J/mol |
| Prefactor | $A_{ds}$ | $1.0275 \times 10^{-7}$ | s$^{-1}$ |
| Viscosity jump | $\Delta\eta_{ds}$ | 2021.20 | Pa s |
| Stress exponent dislocation creep | $n$ | 3.5 | |
| | | | |
| **Grain size evolution parameters** | | | |
| Initial grain size | $\mathcal{R}_0$ | 100.0 | $\mu$m |
| Grain growth exponent | $p$ | 4.5 | |
| Grain surface tension | $\gamma$ | $10^6$ | Pa$\mu$m |
| Activation energy | $E_G$ | $4.14 \times 10^5$ | J/mol |
| Experimental prefactor | $k_0$ | $3.9811 \times 10^6$ | $\mu$m$^p$/s |
| Constant | $\lambda_2$ | 3.5966 | |
| Constant | $\lambda_3$ | 17.81427 | |
| Grain size reset depths | | 2740/660/520/410 | km |
| Grain size after phase transition | $\mathcal{R}_T$ | 5.0 | $\mu$m |
| Damage fraction at 4000 K | $f_{\mathrm{bot}}$ | $10^{-7}$ | |





**Table 2.** List of tested parameters.

| Parameter | Symbol | Value | Units |
|---|---|---|---|
| **Primordial layer** | | | |
| Surface density: primordial | $\rho_{\text{prim}}$ | 3080/3140 | kg/m$^3$ |
| Buoyancy number: primordial | $B_{\text{prim}}$ | 0.0/0.14 | |
| Viscosity factor | $\eta_{\text{prim}}$ | 1/10 | |
| Thickness | D$_{\text{prim}}$ | 200 | km |
| | | | |
| **Model parameter** | | | |
| Yield stress | $\tau_y$ | 10/20/40 | MPa |
| Yield stress gradient | $c_{\tau_y}$ | 0.05/0.1/0.2 | |
| Eruption efficiency | $er$ | 0.5/0.7 | |
| Diffusion creep efficiency: upper mantle | $\chi_{\text{UM}}$ | 0.1/1.0/10.0 | |
| Diffusion creep efficiency: lower mantle | $\chi_{\text{LM}}$ | 0.1/1.0/10.0 | |
| Maximum damage fraction | $f_{\text{top}}$ | $10^{-2}/10^{-3}/10^{-5}$ | |



**Table 3.** All simulations with input parameters and resulting pile properties and surface velocities averaged over the whole simulation period of 4.5 Gyr. $\rho_{prim}$=3080 kg/m³ and $\rho_{prim}$=3140 kg/m³ are the densities of the primordial material at surface and correspond to a buoyancy number of 0.0 and 0.24, respectively.

| No | er | $f_{top}$ | $\chi_{UM}$ | $\chi_{LM}$ | $A_{df} \times 10^{-5}$ [1/s] | $\Delta\eta_{df}$ [Pa s] | $\tau_y$ [MPa] | $\rho_{prim}$ [kg/m³] | $\eta_{prim}$ | $c_{\tau_y}$ | $\langle T \rangle$ [K] | $\langle \rho \rangle$ [kg/m³] | $\langle \eta \rangle \times 10^{22}$ [MPa] | $\langle rheo \rangle$ disl/dif | $\langle \mathcal{R} \rangle$ [µm] | $\langle v_{surf} \rangle$ [cm/yr] |
|---|---|---|---|---|---|---|---|---|---|---|---|---|---|---|---|---|
| 1 | 0.5 | $10^{-2}$ | 1 | 1 | 3.0072 | 11.12 | 20 | 3080 | 1 | 0.1 | 4253.76 | 5564.18 | 5.963 | 0.0904 | 7759 | 27.23 |
| 2 | 0.5 | $10^{-2}$ | 1 | 1 | 3.0072 | 11.12 | 20 | 3080 | 1 | 0.2 | 4337.12 | 5523.78 | 7.465 | 0.1361 | 9171 | 12.66 |
| **3** | **0.5** | $\mathbf{10^{-2}}$ | **1** | **1** | **3.0072** | **11.12** | **20** | **3080** | **10** | **0.1** | **4271.78** | **5544.65** | **70.17** | **0.2520** | **8885** | **20.55** |
| 4 | 0.5 | $10^{-2}$ | 1 | 1 | 3.0072 | 11.12 | 20 | 3080 | 10 | 0.2 | 4349.20 | 5518.59 | 56.91 | 0.3305 | 9159 | 8.52 |
| 5 | 0.5 | $10^{-2}$ | 1 | 1 | 3.0072 | 11.12 | 20 | 3140 | 1 | 0.1 | 4284.40 | 5668.45 | 7.715 | 0.1212 | 8636 | 21.73 |
| 6 | 0.5 | $10^{-2}$ | 1 | 1 | 3.0072 | 11.12 | 20 | 3140 | 1 | 0.2 | 4354.56 | 5644.38 | 7.507 | 0.1167 | 9563 | 7.50 |
| **7** | **0.5** | $\mathbf{10^{-2}}$ | **1** | **1** | **3.0072** | **11.12** | **20** | **3140** | **10** | **0.1** | **4321.82** | **5665.96** | **63.17** | **0.2110** | **8913** | **49.83** |
| 8 | 0.5 | $10^{-2}$ | 1 | 1 | 3.0072 | 11.12 | 20 | 3140 | 10 | 0.2 | 4363.32 | 5651.93 | 68.241 | 0.2779 | 9514 | 15.41 |
| 9 | 0.5 | $10^{-2}$ | 1 | 1 | 3.0072 | 11.12 | 40 | 3080 | 1 | 0.1 | 4304.59 | 5561.21 | 7.900 | 0.1195 | 8367 | 21.64 |
| 10 | 0.5 | $10^{-2}$ | 1 | 1 | 3.0072 | 11.12 | 40 | 3080 | 1 | 0.2 | 4436.71 | 5559.17 | 5.747 | 0.1110 | 9555 | 1.81 |
| 11 | 0.5 | $10^{-2}$ | 1 | 1 | 3.0072 | 11.12 | 40 | 3080 | 10 | 0.1 | 4285.03 | 5566.33 | 55.59 | 0.2303 | 8591 | 28.65 |
| 12 | 0.5 | $10^{-2}$ | 1 | 1 | 3.0072 | 11.12 | 40 | 3080 | 10 | 0.2 | 4380.56 | 5522.77 | 61.72 | 0.3916 | 9456 | 8.19 |
| 13 | 0.5 | $10^{-2}$ | 1 | 1 | 3.0072 | 11.12 | 40 | 3140 | 1 | 0.1 | 4354.50 | 5663.74 | 7.078 | 0.0751 | 9131 | 8.85 |
| 14 | 0.5 | $10^{-2}$ | 1 | 1 | 3.0072 | 11.12 | 40 | 3140 | 1 | 0.2 | 4446.54 | 5668.72 | 6.625 | 0.1287 | 10076 | 1.92 |
| 15 | 0.5 | $10^{-2}$ | 1 | 1 | 3.0072 | 11.12 | 40 | 3140 | 10 | 0.1 | 4283.97 | 5661.72 | 76.27 | 0.1722 | 9094 | 16.27 |
| 16 | 0.5 | $10^{-2}$ | 1 | 1 | 3.0072 | 11.12 | 40 | 3140 | 10 | 0.2 | 4372.87 | 5637.50 | 68.57 | 0.2660 | 9638 | 9.40 |
| 17 | 0.5 | $10^{-2}$ | 0.1 | 0.1 | 2.9920 | 10.99 | 10 | 3140 | 1 | 0.05 | 4379.42 | 5645.40 | 15.52 | 14.8012 | 7950 | 39.40 |
| 18 | 0.5 | $10^{-2}$ | 0.1 | 1 | 2.9920 | 10.99 | 10 | 3140 | 1 | 0.05 | 4362.22 | 5670.66 | 8.348 | 1.5988 | 8090 | 51.09 |
| 19 | 0.5 | $10^{-2}$ | 0.1 | 10 | 2.9920 | 10.99 | 10 | 3140 | 1 | 0.05 | 4377.20 | 5656.91 | 9.189 | 0.3687 | 8036 | 37.87 |
| 20 | 0.5 | $10^{-2}$ | 1 | 0.1 | 3.0072 | 11.12 | 10 | 3140 | 1 | 0.05 | 4336.77 | 5673.99 | 12.47 | 1.5483 | 8487 | 34.40 |
| 21 | 0.5 | $10^{-2}$ | 1 | 1 | 3.0072 | 11.12 | 10 | 3140 | 1 | 0.05 | 4297.63 | 5675.02 | 6.453 | 0.1008 | 8417 | 29.93 |
| 22 | 0.5 | $10^{-2}$ | 1 | 10 | 3.0072 | 11.12 | 10 | 3140 | 1 | 0.05 | 4318.68 | 5676.10 | 4.443 | 0.0331 | 8489 | 42.07 |
| 23 | 0.5 | $10^{-2}$ | 10 | 0.1 | 3.0676 | 11.05 | 10 | 3140 | 1 | 0.05 | 4309.52 | 5667.35 | 18.06 | 0.2590 | 8644 | 8.85 |
| 24 | 0.5 | $10^{-2}$ | 10 | 1 | 3.0676 | 11.05 | 10 | 3140 | 1 | 0.05 | 4282.44 | 5671.95 | 3.959 | 0.0059 | 8310 | 10.74 |
| 25 | 0.5 | $10^{-2}$ | 10 | 10 | 3.0676 | 11.05 | 10 | 3140 | 1 | 0.05 | 4340.99 | 5679.48 | 2.415 | 0.0004 | 8904 | 13.82 |
| 26 | 0.5 | $10^{-2}$ | 0.1 | 0.1 | 2.9920 | 10.99 | 10 | 3140 | 1 | 0.1 | 4453.34 | 5690.35 | 6.350 | 14.1498 | 6797 | 42.46 |
| 27 | 0.5 | $10^{-2}$ | 0.1 | 1 | 2.9920 | 10.99 | 10 | 3140 | 1 | 0.1 | 4276.89 | 5674.71 | 20.72 | 1.4979 | 8655 | 36.14 |
| 28 | 0.5 | $10^{-2}$ | 0.1 | 10 | 2.9920 | 10.99 | 10 | 3140 | 1 | 0.1 | 4291.29 | 5670.48 | 16.90 | 0.3427 | 8616 | 35.74 |
| 29 | 0.5 | $10^{-2}$ | 1 | 0.1 | 3.0072 | 11.12 | 10 | 3140 | 1 | 0.1 | 4302.06 | 5669.04 | 18.15 | 1.7495 | 8675 | 22.21 |
| 30 | 0.5 | $10^{-2}$ | 1 | 1 | 3.0072 | 11.12 | 10 | 3140 | 1 | 0.1 | 4286.22 | 5674.92 | 6.754 | 0.0944 | 8374 | 25.05 |
| 31 | 0.5 | $10^{-2}$ | 1 | 10 | 3.0072 | 11.12 | 10 | 3140 | 1 | 0.1 | 4259.76 | 5678.40 | 4.721 | 0.0086 | 8364 | 31.32 |
| 32 | 0.5 | $10^{-2}$ | 10 | 0.1 | 3.0676 | 11.05 | 10 | 3140 | 1 | 0.1 | 4304.83 | 5666.00 | 15.44 | 0.3429 | 8580 | 13.89 |
| 33 | 0.5 | $10^{-2}$ | 10 | 1 | 3.0676 | 11.05 | 10 | 3140 | 1 | 0.1 | 4447.84 | 5669.78 | 4.264 | 0.0052 | 10021 | 1.66 |
| 34 | 0.5 | $10^{-2}$ | 10 | 10 | 3.0676 | 11.05 | 10 | 3140 | 1 | 0.1 | 4441.97 | 5671.85 | 2.616 | 0.0002 | 10219 | 2.00 |
| 35 | 0.5 | $10^{-2}$ | 0.1 | 0.1 | 2.9920 | 10.99 | 20 | 3140 | 1 | 0.05 | 4366.84 | 5672.23 | 11.56 | 16.3380 | 8371 | 30.38 |
| 36 | 0.5 | $10^{-2}$ | 0.1 | 1 | 2.9920 | 10.99 | 20 | 3140 | 1 | 0.05 | 4328.90 | 5666.94 | 10.75 | 1.0671 | 8472 | 32.59 |


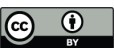

**Table 4.** Table continued. All simulations with input parameters and resulting pile properties and surface velocities averaged over the whole simulation period of 4.5 Gyr. $\rho_{prim}$ corresponds to the density of the primordial material at surface.

| No | er | $f_{top}$ | $\chi_{UM}$ | $\chi_{LM}$ | $A_{df} \times 10^{-5}$ [1/s] | $\Delta\eta_{df}$ [Pa s] | $\tau_y$ [MPa] | $\rho_{prim}$ [kg/m³] | $\eta_{prim}$ | $c_{\tau y}$ | $\langle T \rangle$ [K] | $\langle \rho \rangle$ [kg/m³] | $\langle \eta \rangle \times 10^{22}$ [Pa s] | $\langle rheo \rangle$ disl/dif | $\langle \mathcal{R} \rangle$ [μm] | $\langle v_{surf} \rangle$ [cm/yr] |
|---|---|---|---|---|---|---|---|---|---|---|---|---|---|---|---|---|
| 37 | 0.5 | $10^{-2}$ | 0.1 | 10 | 2.9920 | 10.99 | 20 | 3140 | 1 | 0.05 | 4353.69 | 5642.44 | 20.07 | 0.4082 | 8205 | 35.14 |
| 38 | 0.5 | $10^{-2}$ | 1 | 0.1 | 3.0072 | 11.12 | 20 | 3140 | 1 | 0.05 | 4349.87 | 5598.29 | 7.773 | 0.7399 | 9084 | 9.61 |
| 39 | 0.5 | $10^{-2}$ | 1 | 1 | 3.0072 | 11.12 | 20 | 3140 | 1 | 0.05 | 4343.82 | 5672.49 | 7.425 | 0.0906 | 9314 | 14.03 |
| 40 | 0.5 | $10^{-2}$ | 1 | 10 | 3.0072 | 11.12 | 20 | 3140 | 1 | 0.05 | 4284.55 | 5677.04 | 4.489 | 0.0163 | 8405 | 31.85 |
| 41 | 0.5 | $10^{-2}$ | 10 | 0.1 | 3.0676 | 11.05 | 20 | 3140 | 1 | 0.05 | 4302.21 | 5670.71 | 21.30 | 0.3028 | 8529 | 20.74 |
| 42 | 0.5 | $10^{-2}$ | 10 | 1 | 3.0676 | 11.05 | 20 | 3140 | 1 | 0.05 | 4291.89 | 5666.45 | 19.56 | 0.0254 | 8818 | 13.78 |
| 43 | 0.5 | $10^{-2}$ | 10 | 10 | 3.0676 | 11.05 | 20 | 3140 | 1 | 0.05 | 4358.19 | 5667.40 | 2.822 | 0.0043 | 9156 | 12.95 |
| 44 | 0.5 | $10^{-2}$ | 0.1 | 0.1 | 2.9920 | 10.99 | 20 | 3140 | 1 | 0.1 | 4360.66 | 5659.31 | 11.72 | 14.9736 | 8067 | 54.34 |
| 45 | 0.5 | $10^{-2}$ | 0.1 | 1 | 2.9920 | 10.99 | 20 | 3140 | 1 | 0.1 | 4341.49 | 5670.35 | 8.855 | 1.7508 | 8193 | 32.25 |
| 46 | 0.5 | $10^{-2}$ | 0.1 | 10 | 2.9920 | 10.99 | 20 | 3140 | 1 | 0.1 | 4308.81 | 5671.69 | 19.44 | 0.3240 | 8682 | 24.11 |
| 47 | 0.5 | $10^{-2}$ | 1 | 0.1 | 3.0072 | 11.12 | 20 | 3140 | 1 | 0.1 | 4297.96 | 5665.02 | 17.49 | 1.6447 | 8491 | 19.35 |
| 48 | 0.5 | $10^{-2}$ | 1 | 1 | 3.0072 | 11.12 | 20 | 3140 | 1 | 0.1 | 4295.06 | 5636.09 | 6.203 | 0.1304 | 8282 | 35.84 |
| 49 | 0.5 | $10^{-2}$ | 1 | 10 | 3.0072 | 11.12 | 20 | 3140 | 1 | 0.1 | 4265.95 | 5674.15 | 5.235 | 0.0132 | 8429 | 41.05 |
| 50 | 0.5 | $10^{-2}$ | 10 | 0.1 | 3.0676 | 11.05 | 20 | 3140 | 1 | 0.1 | 4320.23 | 5664.66 | 15.63 | 0.3388 | 8620 | 7.51 |
| 51 | 0.5 | $10^{-2}$ | 10 | 1 | 3.0676 | 11.05 | 20 | 3140 | 1 | 0.1 | 4437.48 | 5669.89 | 4.368 | 0.0055 | 10021 | 1.57 |
| 52 | 0.5 | $10^{-2}$ | 10 | 10 | 3.0676 | 11.05 | 20 | 3140 | 1 | 0.1 | 4440.42 | 5672.04 | 2.515 | 0.0002 | 10091 | 1.88 |
| 53 | 0.5 | $10^{-2}$ | 1 | 1 | 3.0072 | 11.12 | 10 | 3140 | 1 | 0.05 | 4301.89 | 5674.86 | 24.78 | 0.1313 | 8527 | 24.47 |
| 54 | 0.5 | $10^{-3}$ | 1 | 1 | 3.0734 | 8.416 | 10 | 3140 | 1 | 0.05 | 4285.42 | 5661.50 | 12.67 | 0.0610 | 8289 | 25.72 |
| 55 | 0.5 | $10^{-5}$ | 1 | 1 | 3.1166 | 7.632 | 10 | 3140 | 1 | 0.05 | 4352.66 | 5671.68 | 4.869 | 0.0421 | 8952 | 27.10 |
| 56 | 0.5 | $10^{-2}$ | 1 | 1 | 3.0072 | 11.12 | 10 | 3140 | 1 | 0.1 | 4279.22 | 5663.11 | 6.783 | 0.0953 | 8372 | 17.54 |
| 57 | 0.5 | $10^{-3}$ | 1 | 1 | 3.0734 | 8.416 | 10 | 3140 | 1 | 0.1 | 4430.70 | 5671.34 | 5.494 | 0.0546 | 9977 | 4.01 |
| 58 | 0.5 | $10^{-5}$ | 1 | 1 | 3.1166 | 7.632 | 10 | 3140 | 1 | 0.1 | 4428.71 | 5662.55 | 4.793 | 0.0395 | 9595 | 6.22 |
| 59 | 0.5 | $10^{-2}$ | 1 | 1 | 3.0072 | 11.12 | 20 | 3140 | 1 | 0.1 | 4298.10 | 5675.91 | 13.52 | 0.1010 | 8556 | 35.26 |
| 60 | 0.5 | $10^{-3}$ | 1 | 1 | 3.0734 | 8.416 | 20 | 3140 | 1 | 0.05 | 4306.78 | 5675.22 | 4.991 | 0.0503 | 8329 | 23.13 |
| 61 | 0.5 | $10^{-5}$ | 1 | 1 | 3.1166 | 7.632 | 20 | 3140 | 1 | 0.05 | 4409.25 | 5671.76 | 4.446 | 0.0360 | 9239 | 15.71 |
| 62 | 0.5 | $10^{-2}$ | 1 | 1 | 3.0072 | 11.12 | 20 | 3140 | 1 | 0.1 | 4317.56 | 5674.08 | 7.137 | 0.0735 | 8679 | 17.46 |
| 63 | 0.5 | $10^{-3}$ | 1 | 1 | 3.0734 | 8.416 | 20 | 3140 | 1 | 0.1 | 4429.18 | 5671.24 | 5.541 | 0.0563 | 9989 | 3.79 |
| 64 | 0.5 | $10^{-5}$ | 1 | 1 | 3.1166 | 7.632 | 20 | 3140 | 1 | 0.1 | 4417.55 | 5666.28 | 4.995 | 0.0423 | 9515 | 22.18 |
| 65 | 0.7 | $10^{-2}$ | 1 | 1 | 3.0072 | 11.12 | 10 | 3140 | 1 | 0.05 | 4268.21 | 5676.03 | 6.875 | 0.1003 | 8368 | 29.79 |
| 66 | 0.7 | $10^{-2}$ | 1 | 1 | 3.0072 | 11.12 | 10 | 3140 | 1 | 0.1 | 4273.89 | 5673.79 | 7.827 | 0.0645 | 8466 | 47.38 |
| 67 | 0.7 | $10^{-2}$ | 1 | 1 | 3.0072 | 11.12 | 10 | 3140 | 1 | 0.2 | 4390.65 | 5666.79 | 6.952 | 0.0825 | 9589 | 2.68 |
| 68 | 0.7 | $10^{-2}$ | 1 | 1 | 3.0072 | 11.12 | 10 | 3140 | 10 | 0.05 | 4303.81 | 5669.71 | 63.06 | 0.2178 | 8923 | 45.85 |
| 69 | 0.7 | $10^{-2}$ | 1 | 1 | 3.0072 | 11.12 | 10 | 3140 | 10 | 0.1 | 4278.83 | 5664.68 | 77.69 | 0.2334 | 9049 | 30.17 |
| 70 | 0.7 | $10^{-2}$ | 1 | 1 | 3.0072 | 11.12 | 10 | 3140 | 10 | 0.2 | 4372.58 | 5649.61 | 76.46 | 0.2348 | 10128 | 3.07 |
| 71 | 0.7 | $10^{-2}$ | 1 | 1 | 3.0072 | 11.12 | 20 | 3140 | 1 | 0.05 | 4278.81 | 5675.95 | 7.182 | 0.0877 | 8490 | 24.66 |
| **72** | **0.7** | $\mathbf{10^{-2}}$ | **1** | **1** | **3.0072** | **11.12** | **20** | **3140** | **1** | **0.1** | **4293.64** | **5668.26** | **7.602** | **0.0764** | **8668** | **17.09** |





**Table 5.** Table continued. All simulations with input parameters and resulting pile properties and surface velocities averaged over the whole simulation period of 4.5 Gyr. $\rho_{\text{prim}}$ corresponds to the density of the primordial material at surface.

| No | er | $f_{\text{top}}$ | $\chi_{\text{UM}}$ | $\chi_{\text{LM}}$ | $A_{df} \times 10^{-5}$ [1/s] | $\Delta\eta_{df}$ [Pa s] | $\tau_y$ [MPa] | $\rho_{\text{prim}}$ [kg/m³] | $\eta_{\text{prim}}$ | $c_{\tau_y}$ | $\langle T \rangle$ [K] | $\langle \rho \rangle$ [kg/m³] | $\langle \eta \rangle \times 10^{22}$ [Pa s] | $\langle \text{rheo} \rangle$ disl/dif | $\langle \mathcal{R} \rangle$ [$\mu$m] | $\langle v_{\text{surf}} \rangle$ [cm/yr] |
|---|---|---|---|---|---|---|---|---|---|---|---|---|---|---|---|---|
| **73** | **0.7** | **$10^{-2}$** | **1** | **1** | **3.0072** | **11.12** | **20** | **3140** | **1** | **0.2** | **4368.60** | **5666.51** | **6.918** | **0.0797** | **9270** | **2.94** |
| 74 | 0.7 | $10^{-2}$ | 1 | 1 | 3.0072 | 11.12 | 20 | 3140 | 10 | 0.05 | 4294.48 | 5670.78 | 65.58 | 0.2349 | 8940 | 21.67 |
| 75 | 0.7 | $10^{-2}$ | 1 | 1 | 3.0072 | 11.12 | 20 | 3140 | 10 | 0.1 | 4287.19 | 5664.63 | 60.22 | 0.2135 | 8638 | 22.16 |
| 76 | 0.7 | $10^{-2}$ | 1 | 1 | 3.0072 | 11.12 | 20 | 3140 | 10 | 0.2 | 4378.59 | 5629.93 | 61.70 | 0.4641 | 9968 | 9.11 |
| 77 | 0.7 | $10^{-2}$ | 1 | 1 | 3.0072 | 11.12 | 40 | 3140 | 1 | 0.05 | 4277.47 | 5676.69 | 6.894 | 0.0955 | 8389 | 55.24 |
| 78 | 0.7 | $10^{-2}$ | 1 | 1 | 3.0072 | 11.12 | 40 | 3140 | 1 | 0.1 | 4280.98 | 5666.12 | 6.427 | 0.1049 | 8278 | 19.16 |
| 79 | 0.7 | $10^{-2}$ | 1 | 1 | 3.0072 | 11.12 | 40 | 3140 | 1 | 0.2 | 4457.45 | 5668.51 | 7.017 | 0.1050 | 10290 | 1.90 |
| 80 | 0.7 | $10^{-2}$ | 1 | 1 | 3.0072 | 11.12 | 40 | 3140 | 10 | 0.05 | 4257.24 | 5670.53 | 64.58 | 0.2347 | 8523 | 54.44 |
| 81 | 0.7 | $10^{-2}$ | 1 | 1 | 3.0072 | 11.12 | 40 | 3140 | 10 | 0.1 | 4289.34 | 5665.03 | 79.99 | 0.1964 | 9123 | 20.08 |
| 82 | 0.7 | $10^{-2}$ | 1 | 1 | 3.0072 | 11.12 | 40 | 3140 | 10 | 0.2 | 4374.19 | 5637.46 | 62.30 | 0.3482 | 9674 | 16.98 |





**Figure 3.** Example of the dynamics of grain size and viscosity during an overturn (1.58 Gyr), during the mobile lid-phase (2.46 Gyr) and the stagnant lid-phase (4.0 Gyr). The white line outlines the pile, the black line regions with a partial melt percentage higher than 50% (viscosity and grain size in these regions only refer to the solid). On the bottom, the evolving distribution of basalt is displayed. Outlined in white is again the detected pile. In red this time, the regions with partial melt. It can be observed that basalt does not mix with the pile but pushes it aside.





## 3.2 Pile averages

In this section we examine the time-dependent dynamics and properties of the detected piles in detail. We find that the overall pile dynamics and behaviour of the average properties mainly depend on different convection regimes throughout the run time. Therefore, the results are described in light of different tectonic regimes. We differentiate between stagnant lid phase, plate tectonic-like/mobile-lid phase and overturn events.

We exemplary show one simulation and all average pile properties to present their evolution and interaction (additional figures and observations are in the appendix). Pile averages of grain size, stress, strain rate, viscosity, temperature, density and rheology, and the surface velocity are plotted over time (figure 4). The model is the same presented in the prior section. The primordial material of the simulation has the same viscosity and mechanical properties as basalt and a buoyancy number of $B_{\mathrm{prim}} = 0.14$, the yield stress in the simulation is 20 MPa, the yield stress gradient 0.1 and the eruption efficiency 0.7 (model No 72 in table 5). This simulation shows different types of convection regimes: two stagnant lid-phases (up to 1.5 Gyr & after 3.5 Gyr), overturn events (at 1.5 Gyr & at 3.2 & 3.4 Gyr) and a mobile lid-phase between 2.0 Gyr and 3.2 Gyr (figure 4). The convection regimes are differentiated by plate velocity, where 1 cm/yr is the border between mobile and stagnant lid.

### 3.2.1 Stagnant lid-phase

During the first stagnant lid phase (until 1.5 Gyr), grain size and viscosity of the pile both increase and the pile dominantly deforms in diffusion creep. Grain sizes vary between 6000 and 10000 $\mu$m (excluding the initiation phase) and viscosity between $10^{22}$ and $8 \times 10^{22}$ Pa s. The calculated equilibrium grain size plotted in figure 3 is very large during this stage, because stresses are low.

Strain rate, stress and surface velocity decrease after the initiation of the simulation. The minimum strain rate right before the overturn event is $8 \times 10^{-17} \, s^{-1}$ and the minimum stress $5 \times 10^6$ Pa. Surface velocity strongly decreases to less than $10^{-3}$ cm/yr.

Initially, pile average temperature also starts to decrease, but then stays constant which can be attributed to the development of thick crust during the stagnant lid phase. This prevents the Earth to cool down more. Average temperature of the pile is 4400 K. The average density of the pile starts off with the value imposed for the primordial layer and decreases slightly to 5680 $kg/m^3$ until the overturn event occurs at 1.5 Gyr.

During the second stagnant lid phase (3.5-4.5 Gyr) all pile properties recover and grain size, viscosity and density reach values that are higher than during the mobile lid-phase of the simulation. The surface velocity is not as low as during the first stagnant lid phase, but rather close to the mobile lid phase. Accordingly, the average stress of the pile is a higher than during the first stagnant lid phase. The small variations in surface velocity are reflected in the average stress, strain rate and rheology of the pile. The pile temperature can further decrease during the second stagnant lid phase because there exists still some movement at the surface.





### 3.2.2 Episodic overturn/Resurfacing phase

An overturn event (at 1.5 Gyr or 3.2 Gyr) is marked by a very high surface velocity because a lot of cold lithosphere simultaneously moves down into the mantle. Hence, the resurfacing is associated with a sudden increase and peak in the average strain rate and stress of the pile material due to the push of the downwelling lithospheric material. The high fluctuations of stress
lead to a very low equilibrium grain size which resets the grain size in the piles during the overturn and downwellings events. Following the diminished grain size, the viscosity decreases as well.

The rheology is dislocation-dominated during the high stress phase, and then quickly returns to diffusion-dominated once the grains are small and have not had time to grow back yet. Since the period of high stress and strain rate is short, grain size and viscosity quickly recover and return to the values prior to the overturn event (see 3.2.4).

Density decreases during the resurfacing as well which can be explained by the relocation of pile material. The pile moves away from the CMB where the cold and stiff previous lithosphere accumulates. Therefore, pile material rises up higher where density is lower. Similar to viscosity and grain size, density recovers after the pile re-settles along a wider area of the CMB.

### 3.2.3 Plate tectonic-like/Mobile lid-phase

During the mobile lid-phase, stress, strain rate, rheology, and surface velocity show a lot of variations. The pile average viscosity
and grain size follow the variations of strain rate and stress, but pile density barely reflects the other properties' variations during the mobile lid-phase. Deformation of the pile is mainly performed in diffusion creep, but with a higher component of dislocation creep than during the stagnant lid-phase. The average pile temperature continuously decreases during the mobile lid-phase because of the absent of an insulating thick lithosphere at surface.

### 3.2.4 Pile recovery time and self-regulating effect

We observe, that in the end of the simulations average properties are all alike, independent of the convection regime and convection history. This is because average properties quickly return to former values ('recovery') after fluctuations due to downwellings or episodic overturns. We call this the 'self-regulating effect' and observe it for all properties (excluding temperature).

The time window of recovery depends, on the one hand, on the vigorousness of the convection (density, stress) and, on the
other hand, on the grains' drive to reach the equilibrium grain size (figure 3, top). Figures 3 and 4 illustrate how fast the piles' grain size and other properties recover after one overturn event. We call this the recovery time $t_{rec}$ of the piles. Specifically for grain size, it can be computed by reformulating the grain growth term to

$$t_{\mathcal{R},\text{rec}} = \frac{\mathcal{R}^p}{G}.$$
(19)

We find the grain size-recovery time to be approximately 420 Myr for a temperature of 4400 K and an estimated recovered
grain size of 9000 $\mu$m. This result relates to the plotted grain size in figure 3.



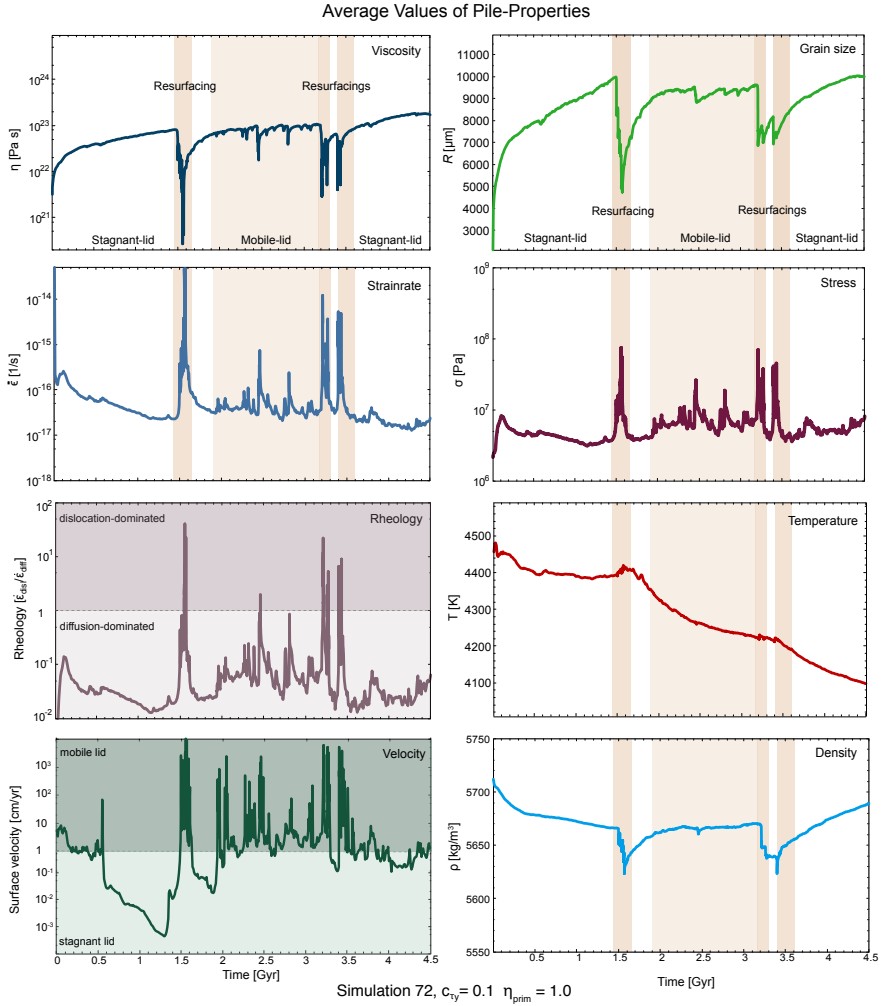

**Figure 4.** Average pile properties for the whole simulation time and the surface velocity (bottom left) of the simulation to classify the convection regime. Low viscosity, low grain size, high stress and strain rate and dislocation-dominated rheology are correlated and occur during overturn events.

### 3.2.5 Dependency of Pile Properties on input Parameters

In order to estimate the importance of each input parameters on the effective properties of the thermo-chemical piles, we perform empirical regressions of the time and space averages reported in table 5. For temperature and density we use an additive form, since their variations are rather small. Grain size, viscosity and rheology are fitted with a power law equation.

Since we use spatial and temporal averages, we can only report first order correlations. The input parameters that are found to be important are printed in bold characters.





**Table 6.** First-order regressions of pile spatial and temporal averages. Temperature and density are fitted with an additive form as their variations are small. Viscosity, rheology and grain size are fitted with a power law equation.

| Regression $= a_0 + a_1\, er + a_2 \log\left(\frac{f_{top}}{10^{-3}}\right) + a_3 \log\left(\chi_{UM}\right) + a_4 \log\left(\chi_{LM}\right) + a_5 \tau_y + a_6 \frac{\rho_{\mathrm{prim}}-3110}{30} + a_7 \log\left(\eta_{\mathrm{prim}}\right) + a_8 \frac{c_{\tau_y}-0.1}{0.1}$ | | | | | | | | | |
|---|---|---|---|---|---|---|---|---|---|
| | $a_0$ | $a_1$ | $a_2$ | $a_3$ | $a_4$ | $a_5$ | $a_6$ | $a_7$ | $a_8$ | Error |
| Temperature | 4413.74 | **-141.48** | -25.56 | 2.891 | -2.092 | 3.218 | 21.15 | -21.04 | **62.26** | 1.01 % |
| Density | 5594.84 | **31.28** | -0.266 | 1.952 | 4.053 | 0.0484 | **54.92** | -10.51 | -10.40 | 0.22 % |

| Regression $= a_0 \left(\frac{er}{0.6}\right)^{a_1} \left(\frac{f_{top}}{10^{-3}}\right)^{a_2} \chi_{UM}^{a_3} \chi_{LM}^{a_4} \left(\frac{\tau_y}{2\cdot 10^7}\right)^{a_5} \left(\frac{\rho_{\mathrm{prim}}}{3110}\right)^{a_6} \eta_{\mathrm{prim}}^{a_7} \left(\frac{c_{\tau_y}}{0.1}\right)^{a_8}$ | | | | | | | | | |
|---|---|---|---|---|---|---|---|---|---|
| | $a_0$ | $a_1$ | $a_2$ | $a_3$ | $a_4$ | $a_5$ | $a_6$ | $a_7$ | $a_8$ | Error |
| Viscosity | $5.93\cdot 10^{22}$ | 0.151 | 0.0471 | -0.140 | -0.150 | 0.0502 | 2.305 | **0.988** | -0.0264 | 29.96 % |
| Rheology | 0.0739 | 0.0986 | 0.113 | **-1.282** | **-1.081** | $-2.51\cdot 10^{-7}$ | -12.21 | **0.423** | 0.0696 | 36.96 % |
| Grain size | $8.90\cdot 10^3$ | $-9.31\cdot 10^{-6}$ | -0.0131 | 0.0250 | 0.0122 | 0.0180 | 2.731 | 0.0131 | 0.0779 | 4.82 % |

We observe that the pile-temperature mostly depends on the eruption efficiency and the yield stress gradient (table 5). If the eruption efficiency is changed from only intrusive to completely extrusive, the temperature of the pile will decrease. This behaviour can be explained with more cold downwelling basalt in case of a completely extrusive regime. If the yield stress
gradient increases by 0.1, the pile average temperature rises. Other variables do not significantly influence the pile temperature. The error of around 1% on temperature is relatively high, but we need to consider that we perform these regressions on temporal and spatial averages.

Density and viscosity of the pile mainly depend on the input density and viscosity, respectively. Errors are low for both density (0.22%) and viscosity (≈30%), taking into account the logarithmic behaviour of viscosity. The average rheology of the
pile is mainly affected by the prescribed effectiveness of diffusion creep in the upper and lower mantle ($\chi_{UM}$ and $\chi_{LM}$), and to a lower extent by the prefactor of the initial viscosity of the pile.

Interestingly, the average grain size does not depend on any of the input parameter. All exponents are very small and the error with 4% is low (table (5)), meaning the regression fits the behaviour of grain size well. This result underlines the self-regulating behaviour of grain size evolution in an evolutionary convection model.

## 3.3 1D-profiles

In this section we report detailed observations on the differences between pile and ambient mantle properties, focusing on viscosity, grain size, temperature and rheology during different tectonic phases. To investigate how these properties evolve with time, we again show profiles inside and outside the pile for five different time steps, using model No 72.

We first present some general observations of how the investigated properties vary within the ambient mantle and the piles.





### 3.3.1 Grain size - General trend

Grain size is very small in the lithosphere and quickly increases to sizes around 1000 $\mu$m in the upper mantle. Differences between different time steps are negligible (figure 5). Below 660 km, grains become larger and the differences in-between time steps increase as well. Inside the pile, grains are larger than in the ambient mantle. The post-perovskite transition at 2740 km leads to a reduction in grain size within the piles as well as within the ambient mantle. However, grain size quickly grows after

passing the transition and a final grain size of around 10000 $\mu$m is reached at the CMB.

### 3.3.2 Viscosity - General trend

Next, we investigate how viscosity changes with time and how ambient mantle-viscosity differs from pile-viscosity. We observe that all sub-figures show a similar behaviour. Generally, the viscosity is very high in the crust, then decreases up to the 660 km boarder where it instantly rises to a value of around $10^{23}$ Pa s. This value remains approximately constant until the

post-perovskite phase transition is reached. There, the viscosity increases rapidly up to the core-mantle boundary. Different time snaps do not display a significantly different behaviour. An exception are viscosities very close to the CMB. Likely, the variations arise due to the amount of subducted material accumulated at that certain time snap at the CMB. Within the described general trend there are some variations, depending on the set of input parameters. These variations are described below.

### 3.3.3 Rheology - General trend

At all time steps, the lithosphere deforms in diffusion creep, the upper mantle is diffusion creep-dominated but shows also a strong component of dislocation creep. The mid- and lower ambient mantle deform in diffusion creep. At the CMB, deformation mechanisms vary strongly, from completely diffusion-dominated creep to dislocation creep-governed deformation. Piles deform with diffusion creep, whereas the lowermost ambient mantle is governed by dislocation creep because it is slightly warmer. The grain size-reset at the post-perovskite transition (2740 km) is responsible for the increase in diffusion creep-

accommodated deformation within the pile.

### 3.3.4 Temperature - General Trend

The temperature increases rapidly in the upper 400 km, followed by a nearly steady temperature and a second pronounced increase in the lowermost mantle from around 2500 km up to the CMB.

### 3.3.5 Convection regime dependence

During the initial stagnant lid-phase, grains are generally still relatively small, and viscosity in the ambient mantle is high which coincides with the lower temperature. During this phase, the deformation is strongly dominated by diffusion creep. Right before 1.5 Gyr, a resurfacing starts. At 1.5 Gyr, a slab has already subducted and the rest of the lithosphere follows shortly after. The deformation mechanism has a higher component of dislocation creep due to stress induced by the downwelling basaltic material and the large grain size which reaches its maximum at this time step. Because of the latter, viscosity is





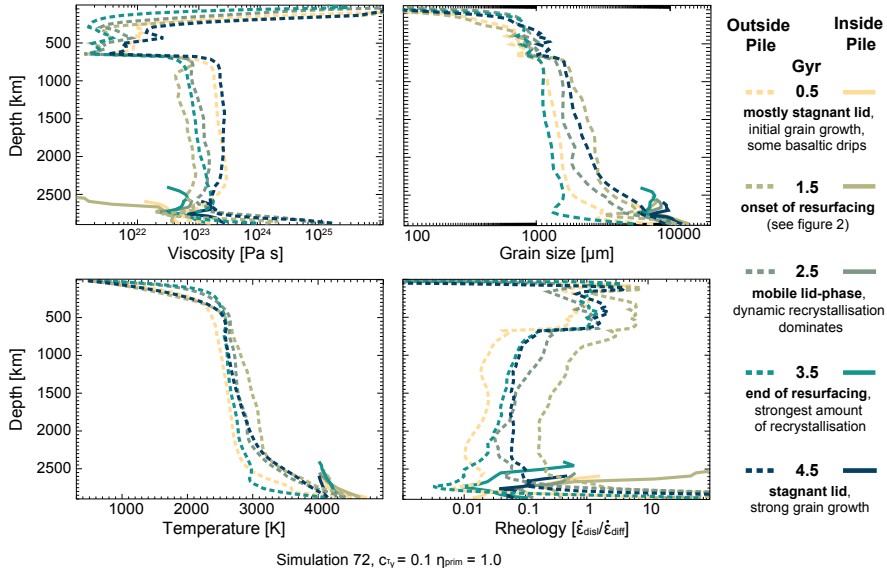

**Figure 5.** 1D-profiles of grain size, viscosity, temperature and rheology through the whole model domain (model No 72). The dashed lines show the average values of crust and ambient mantle for five time steps, the solid curves show average properties within the pile for the same time steps. Convection regime descriptions are provided in the legend.

high, although temperature also reaches the maximum. At time step 2.5 Gyr, the convection regime is plate-tectonic-like with constant downwellings inducing constant stress. This results in a decrease in grain size, viscosity and temperature. Following the recrystallization of grains, the deformation is strongly dominated by diffusion creep. The profiles plotted for 3.5 Gyr show the deformation regime, grain size, viscosity and temperature right at the end of two resurfacing events. Accordingly, the grains have strongly recrystallized which is succeed by a decrease in viscosity. The rheology also shows, by a slightly higher component of diffusion creep than before, that grains are smaller than at 2.5 Ga, and that the constant stress has stopped. At 4.5 Gyr, the model has been in stagnant lid for around 1 Gyr which leads to a increase in temperature and strong grain growth. Viscosity increases a lot, accordingly. At the same time, dislocation creep gets slightly more important again, but the deformation is still governed by diffusion creep.

### 3.4 Influence of Grain size and Temperature on the Viscosity of the Pile and the Mantle

Investigating average values for temperature, grain size and viscosity inside the pile helps us understand the relative importance of grain size and temperature on the viscosity of the pile. Exemplary, we look at two cases (No 3 and No 7 in table 5). The two runs use identical parameters except for the density of the primordial layer: $B_{prim}$ =0.0 in the simulation shown on the left side (No 3), and ($B_{prim}$ =0.14) in the model shown on the right side (No 7) in figure 6.

Figure 6 demonstrates that grain size and viscosity evolution are correlated in the pile. Both, a) and b) show an increase in viscosity when grains grow. However, grains only start growing after viscosity has already increased e.g. after a downwelling (figure 6 a)). This implies that viscosity does not solely depend on grain size. We additionally observe a correlation between





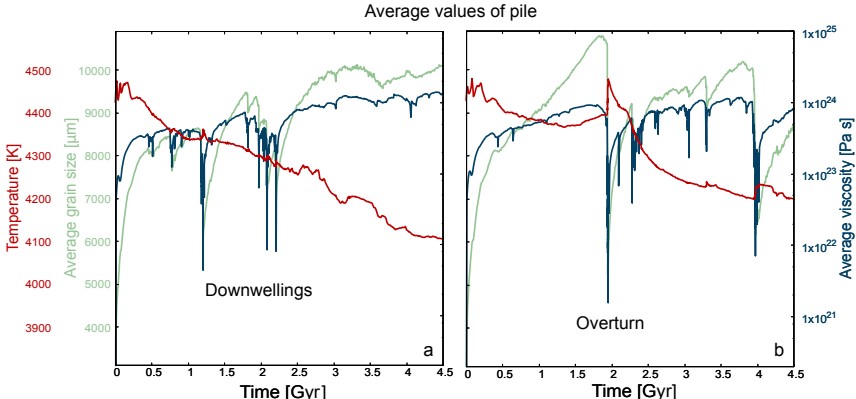

**Figure 6.** Average values for temperature, grain size and viscosity inside the pile. a) Simulation 3 shows the effect of several strong down-welling events in the early stages of the evolution. b) Simulation 7 displays the effect of two overturn events, intermittent by a mobile lid-phase. All properties are plotted in the same graph to emphasize the correlation between grain size and viscosity development and the anti-correlation of temperature evolution.

rising temperature and decreasing viscosity in the pile, e.g. after the overturn event or during the first 0.5 Gyr (figure 6 b). The general trend of decreasing temperature is reconcilable with the overall increase in viscosity. We also find, that grain size and temperature are anti-correlated, although one might expect that grains stop or slow down their growth when temperature

decreases. The observed anti-correlation is explicable with several arguments: although the overall temperature inside the pile decreases, the actual temperature inside the pile is high enough for grains to grow. Secondly, grain growth does mainly depend on the absence of stress or strain rate. If the strain rate within the pile is small, grains will grow because the damage term is small (equation 12). From the above described findings we conclude that both pile-grain size and pile-temperature buffer the development of pile-viscosity in opposite directions in our simulations.

In figure 7 we present 1D-profiles for five different time steps during the model evolution (simulation No 73). The 1D-profiles show averaged values for each depth inside (solid line) and outside (dashed line) the pile. Temperature and grain size in the ambient mantle steadily increase with time, whereas viscosity decreases. The very low viscosity of the ambient mantle at 1.5 Gyr can be explained with a large downwelling occurring right before 1.5 Gyr which leads to high stresses and strain rates, and accumulates along the CMB. The same downwelling also explains why the grain size has not increased a lot until 1.5 Gyr

and why the grain size is very low along the CMB. The high viscosity close to the CMB at time steps 2.5 Gyr and 3.5 Gyr can be attributed to the accumulation of stiff, subducted material from previous downwellings and resurfacing events. Although the viscosity of the presented simulation decreases with time, models employing a purely temperature-dependent viscosity have a much stronger decrease. By using the average temperatures for a depth of 1500 km at time steps 0.5 Gyr and 4.5 Gyr, we receive a viscosity ratio of

$$\frac{\eta_{T=2600}}{\eta_{T=3200}} = \exp\left[\frac{PV + E}{R}\left(\frac{1}{2600} - \frac{1}{3200}\right)\right] \approx 25.8 \tag{20}$$





using $P = 50$ MPa, $E = 3.75 \times 10^5$ J/mol and $V = 5.5 \times 10^{-7}$ m$^3$/mol and $R = 8.314$ J·K$^{-1}$mol$^{-1}$. With a grain size-dependent viscosity, the viscosity ratio is only $\eta_{\mathcal{R}}(T = 2600)/\eta_{\mathcal{R}}(T = 3200) \approx 2.8$.

From figure 7 we can conclude that in the ambient mantle, grain size and temperature are correlated, and, on the other hand, grain size evolution strongly decreases the effective temperature-dependence of the viscosity. This is the opposite behaviour

of what has been shown in figure 6 for average pile properties. However, the 1D-profiles through pile material in figure 7 support the results presented in figure 6. Hence, we infer, that for the chosen parameters, temperature dominates the viscosity evolution in the ambient mantle, and grain size regulates the viscosity development in the pile. The reason for the small effect of temperature on pile-viscosity is that the pile buffers the core temperature and thus, pile-temperature stays nearly constant over the whole evolution (it varies only 300 K).

## 4   Discussion

### 4.1   Grain size in thermo-chemical Piles and ambient Mantle

Our simulations show that deformation in the lower mantle as well as in thermo-chemical piles is mainly accommodated by diffusion creep. Exceptions during phases of overturn and intense downwelling events result in dislocation creep-dominated deformation or an even contribution of diffusion and dislocation creep in the piles. During these events, the lower mantle

deforms mainly in dislocation creep in regions adjacent to the downwelling. These observations are very similar to findings by McNamara et al. (2002) who also used a composite rheology, though without specifically considering grain size evolution. Although there exists a surprisingly good agreement between our and their results, we observe a different deformation mechanism along the CMB. Whereas McNamara et al. (2002) find diffusion creep to dominate deformation, our simulations rather suggest a slight domination by dislocation creep. However, hypotheses featuring strongly dislocation creep-governed deforma-

tion due to a large grain size because of high temperatures along the CMB (Dannberg et al., 2017) cannot be confirmed. The in some parts of the D"-layer observed anisotropy (Garnero, 2000; Kendall and Silver, 1996), specifically in regions of high stress (Karato, 1998) can be perfectly explained by regionally occurring dislocation creep due to downwelling-induced high stress as has been proposed by Karato (1998).

As noted by (Dannberg et al., 2017), LLSVPs are potential regions for large grain size as the stability of LLSVPs and the

high temperature gives grains the right conditions to grow. However, we find that the size of the grains is limited and reaches an equilibrium grain size which is not very different from the grain size in the ambient mantle (figure 5). Therefore, it is difficult to explain a possible higher stiffness of LLSVPs with large grain size.

As Ranalli and Fischer (1984) mention it is impossible to know the grain size in the lower mantle. Therefore, geodynamic studies, in accordance with mineral physics studies, can provide an estimate of the grain size and are of great relevance to

understand the viscosity and dynamics of the deep Earth. The average grain size we find in the lower mantle is on the order of 2000 to 7000 $\mu m$, increasing with depth and time (in piles generally higher) and could, in the future, be compared to similar geodynamic studies, using the same or different grain size evolution equations. Opposite to thoughts mentioned by Ranalli

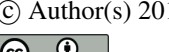



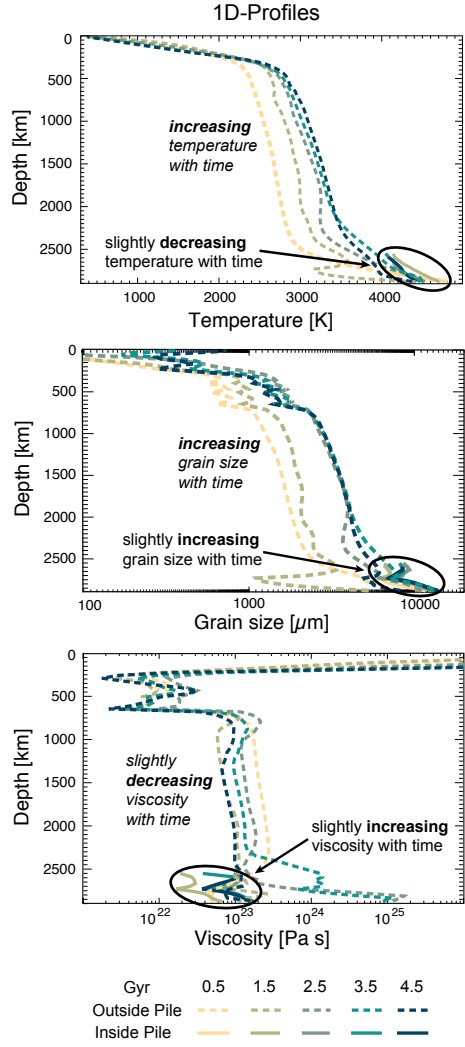

**Figure 7.** 1D-profiles through the whole model domain of simulation 73. The dashed lines show the average values of crust and ambient mantle for five time steps of the model evolution. The darker the color, the later the time step. The solid curves show average properties within the pile for the same time steps. Top: Temperature, middle: Grain size, bottom: Viscosity.

and Fischer (1984), we find that even with a large grain size of up to 7000 $\mu m$ the lower mantle can deform by Newtonian-dominated deformation, and is not necessarily non-linear.

### 4.2 Recovery Time in the Earth


If we assume that in the Earth stresses are generally higher than in the presented model because of continuous subduction, the equilibrium grain size and the recovery time for grain size would be smaller and shorter, respectively. A rough estimate for the equilibrium grain size in the Earth can be calculated by using the relations $\dot{\epsilon}_{\mathrm{Earth}} = v_{\mathrm{plate}}/D_{\mathrm{mantle}}$ and $\tau_{\mathrm{Earth}} = 2\eta_{\mathrm{Earth}}\dot{\epsilon}_{\mathrm{Earth}}$, where



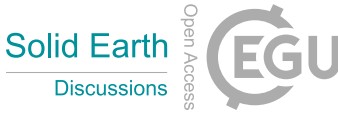

use $v_{\text{plate}} = 3$ cm/yr as the plate velocity at surface, $D_{\text{mantle}} = 3000$ km as the thickness of the Earth's mantle and $\eta_{\text{Earth}} = 5 \times$
$10^{22}$ Pa s as the viscosity in the lower mantle. This results in a strain rate of $\dot{\epsilon}_{\text{Earth}} \approx 3 \times 10^{-16}$ s$^{-1}$ and stress of $\tau_{\text{Earth}} \approx 30$ MPa
which leads to an average equilibrium grain size of around 4000 $\mu$m for Earth's piles (figure 3). The recovery time for this
equilibrium grain size of 4000 $\mu$m would be on the order of 215 Myr, when assuming a temperature of 3500 K inside the piles.
However, the recovery grain size of the pile will probably be smaller than the equilibrium grain size, similar to the observation
shown in figure 3 for the pile in our simulations. Hence, if we instead assume a recovery grain size of only 3000 $\mu$m, we receive
a much shorter recovery time of 50 Myr. Since the recovery time equation (equation 19) is very sensitive to both grain size and
temperature, the recovery time of thermo-chemical piles in the Earth might vary a lot, depending on the temperature and the
deformation history of the pile.

### 4.3   Spatial Distribution of Piles

Our results contribute to the ongoing debate about whether piles are intrinsically stable features which spatially determine
subduction zones, or are rather defined by subducting slabs themselves. Within the parameter range we studied, we observe
that downgoing slabs are responsible for the spatial distribution of piles and their morphology, as has been noted in previous
studies by (e.g. McNamara and Zhong, 2004, 2005). However, unlike findings by McNamara and Zhong (2004), we do not see
a difference in pile morphology when a viscosity contrast between pile-material and ambient mantle is introduced. We further
do not find that grain size assists the stabilisation of thermochemical piles by increasing the resistance towards downgoing
slabs. On the opposite, we note that piles are strong as long as they are not exposed to stress, but weak when slabs exert stress
on the piles. This behaviour can be attributed to the non-Newtonian rheology in the composite rheology formulation.

Our thermo-chemical piles are also not surrounded by plume generation zones (PGZ), as suggested by Burke et al. (2008),
but plumes rise directly from the piles as well as from their margins. Their additional conclusion, that LLVPs and LSVPs
(in geodynamic modelling referred to as thermo-chemical piles) have been stable in time, because the PGZ have been stable
according to Large Igneous Province (LIP) data, can and cannot be confirmed with our models. In our simulations, even though
the piles are negatively buoyant and/or more viscous than the surrounding ambient mantle, downwellings (subduction zones)
govern the piles' spatial distribution and not the other way round. Hence, if there are no strong downwelling events disturbing
the location of the piles, we can receive piles stable for at least 300 Myr. However, since we do not employ realistic plate
velocities we cannot draw conclusions about the actual stability and spatial distribution of LLSVPs, particularly since our
models are only two-dimensional.

### 4.4   Viscosity in thermo-chemical Piles and ambient Mantle

Our results show that grain size has a great impact on the viscosity in numerical convection models. Similar to results by
Dannberg et al. (2017), we find strong lateral variations in grain size and resulting viscosity in our simulations. Overturn events
lead to a distinct 'bimodal' behaviour where half of the spherical annulus shows a distinct decrease in viscosity and smaller
grain size than the other half (figure 3, 1.58 Gyr), where downgoing slabs are surrounded by lower grain size, high strain rate
and viscosity reduction. This finding agrees well with what Dannberg et al. (2017) reported. However, in times without any



particular downwelling event we do not observe very strong lateral viscosity variations in the lower mantle. Larger grain size in plumes does not affect the viscosity of the plumes a lot, and viscosity is relatively uniform in the whole lower mantle ($10^{23}$ to $10^{24}$ Pa s). Thus, the suggestion that higher temperatures in plumes might result in higher viscosity due to larger grains

(Solomatov and Moresi, 1996; Karato and Rubie, 1997; Solomatov et al., 2002; Korenaga, 2005) cannot be supported with our simulations, but is probable when different grain growth parameters would be used.

We further observe that due to the fast recovery of decreased grain size, viscosity quickly reaches values prior to any subduction or overturn event. Although we observe this self-regulating effect specifically for piles, we propose that the whole mantle might behave in a similar way. This proposition is supported by the observation that the viscosity variations with time

are much smaller when using a composite, grain size-dependent viscosity than when using a simple Arrhenius-type viscosity formulation. If the self-regulating effect can also be observed for the whole mantle, the recovery time of grain size could for example be calculated for regions affected by subduction and provide information on healing and deformation recovery.

## 5 Conclusion

Our results demonstrate that thermochemical piles mainly deform in diffusion creep. During downwelling and overturn events,

dislocation creep-accommodated deformation gains importance and can be, but is not necessarily, the dominant deformation mechanism. The spatial distribution of piles depends on the location of subducting slabs and downwelling material. The slightly larger pile-grain size compared to the ambient mantle does not lead to stiff features which are able to dominate the dynamics of the lowermost mantle. Once piles are exposed to stress, they are weak features that are swept around the CMB. This behaviour can be explained by the non-Newtonian rheology with which piles deform. Properties of the piles, such as density, viscosity,

strain rate, stress or grain size are self-regulating, meaning after a significant downwelling/resurfacing the values quickly recover to values prior to the event affecting the pile.

Although in our simulations dislocation creep seldom occurs in the lower mantle, we see its association with downwellings. If this information is transferred to the Earth, we can infer that due to continuous subduction, higher stresses in the Earth exist leading to more dislocation creep, which in turn could explain long-lasting seismic anisotropy in the lowermost mantle.

In our models we find a relatively uniform viscosity in both upper and lower mantle, unless large overturn events occur. Viscosity of hot plumes and thermo-chemical piles do not significantly differ from ambient mantle viscosity. On the other hand, downgoing slabs display a much larger viscosity, even when reaching the CMB. Overall, our results suggest that viscosity depends more on grain size than on temperature, specifically when constant stress due to downwellings and resurfacing events is present. Our results further demonstrate that the viscosity change over time is considerably smaller in simulations using a

grain size-dependent viscosity than in models employing an Arrhenius-type viscosity. These findings let us conclude that grain size is important to consider in the viscosity formulation of evolutionary convection models.





## Appendix A:  Summary of Pile Averages of all Simulations

The average properties of the piles (grain size, viscosity, density, temperature) are plotted against time and are split into two categories: all simulations that show a mostly continuous mechanical behaviour are displayed in green, the models showing
mostly episodic dynamics in blue. Since the results of all simulations overlap, we only present some specific examples and color the area in which a certain behaviour (episodic or continuous) occurs in the respective color. Whereas one single simulation can show one or more tectonic regimes in series, it is always only classified as either continuous or episodic, because the classification is, unlike the tectonic regimes, a summary of the whole run time instead of specific phases during the run.

### A1   Pile Viscosity

In figure A1, the average viscosities of the piles are plotted against time. We observe that the viscosity inside the piles increases from its initial value to relatively steady values, reached around 1.5 Gyr. Although the viscosity undergoes fluctuations the viscosity quickly reaches the prior value again so that the final viscosity in the end of the runtime is approximately the same as it was around 1.5 Gyr. Several simulations show a constant average pile viscosity (green solid line). Some simulations we

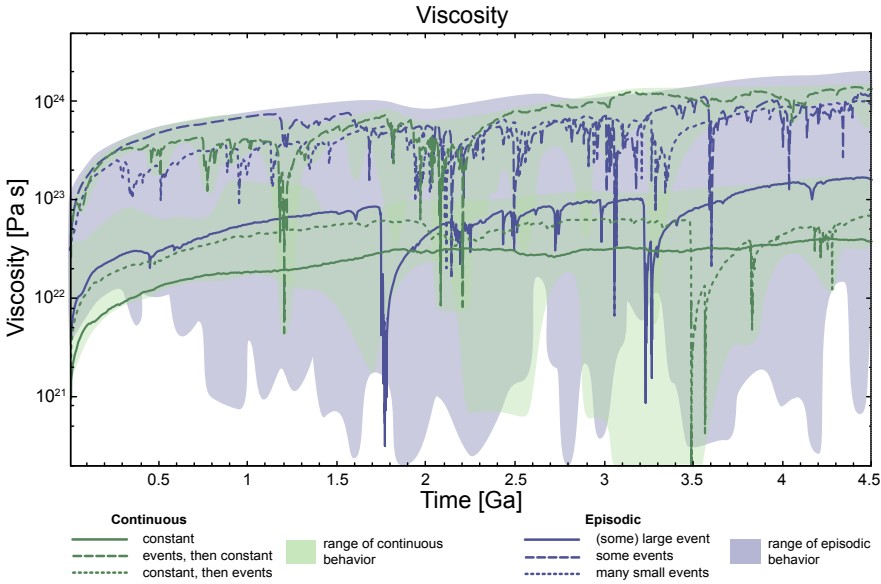

**Figure A1.** Average viscosities of the detected piles plotted over time. Two different behaviours are distinguished: a continuous trend (green) and an episodic trend (blue). Sub-categories are identified by different line types. Background colors show the regions in which certain trends occur.

classify as continuous although they show some events (small downwellings). Either these simulations are constant during the
first half (mostly these models have a higher yield stress), or during the second half of the simulation. These cases either start off with downwellings and dripping and end with a stagnant lid or the other way round. These models are marked by a stagnant





lid with some dripping and delamination at the bottom of the lithosphere. In most cases these models have a high yield stress, but not exclusively, wherefore we cannot state that a high yield stress always leads to a stagnant lid scenario.




## A2 Pile Grain size

The average grain size in the piles shows great similarities with the average pile-viscosity. Again, we distinguish between two different behaviours, where green marks the continuous trends and blue stands for episodic trends (figure A2). The different line types represent sub-categories and the coloured background marks the region in which the associated behaviour occurs. In all simulations we observe an increase in the average pile-grain size until at most 2 Gyr. Grain sizes vary between 7000 and 12000 $\mu$m, excluding sudden drops in grain size linked to overturns. In the end of the runtime the spectrum of grain size is the

same as it already is at 1.5 Gyr. Although large variations in grain size are visible, the grain size quickly grows back to the average 'background'-value from prior to the decrease (self-regulation of grain size). Generally, grain size is larger when fewer downwelling- and overturn-events occur. This is explicable with the lower amount of stress acting on the piles and hence, less grain damage. Details on the interconnection between viscosity, grain size and temperature are given in section 3.4.

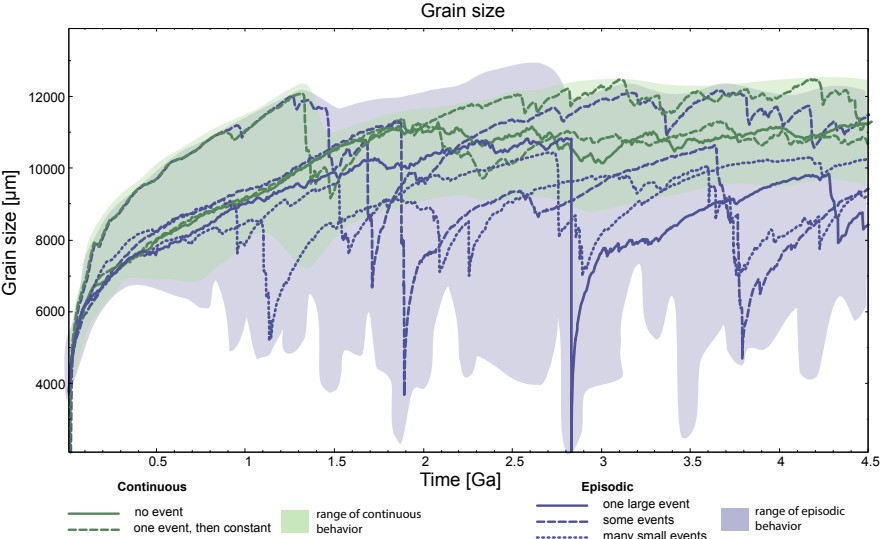

**Figure A2.** Average grain sizes of the detected piles plotted over time. Two different behaviours are shown: continuous trend (green) and episodic trend (blue). Sub-categories are identified by different line types. Background colors show the regions in which certain trends occur.





## A3    Pile Density

We also report the average density of the piles in our simulations. Most of our simulations employ a surface density of 3140 kg/m$^3$ for primordial material ($B_{\text{prim}} = 0.14$). The two lines in the lower part of the figure refer to piles with a surface density of 3080 kg/m$^3$ ($B_{\text{prim}} = 0.0$). Since our piles are allowed to incorporate basalt from downwellings as well the actual buoyancy ratio can slightly deviate from the just named value. However, since we observe that subducted basalt basically does not mix with primordial material (figure 3), the chance of having basalt incorporated in the piles is relatively low.

The density of the piles highly depends on the input density, and the final average density is similar to the input density. We observe an overall slight decrease of density of 50 to 100 kg/m$^3$ between the initial and final average pile density. During the simulation there exist much larger fluctuations, associated with downwellings and overturns that push pile material aside and change the pile structure and location. We can again identify episodic an continuous trends, as observed for grain size and viscosity, though we also find a third trend which is defined by a strong, but slow decrease and increase. Opposite to episodic

behaviour, this last category does not show sudden drops in density, but rather steady decreases and increases that mirror each other. This behaviour can be explained with a slow upward movement of the pile during an overturn event leading to an overall decrease in pile-density, and the slow recovery when the pile sinks back and again spreads out along the CMB. These cases only have one overturn event.

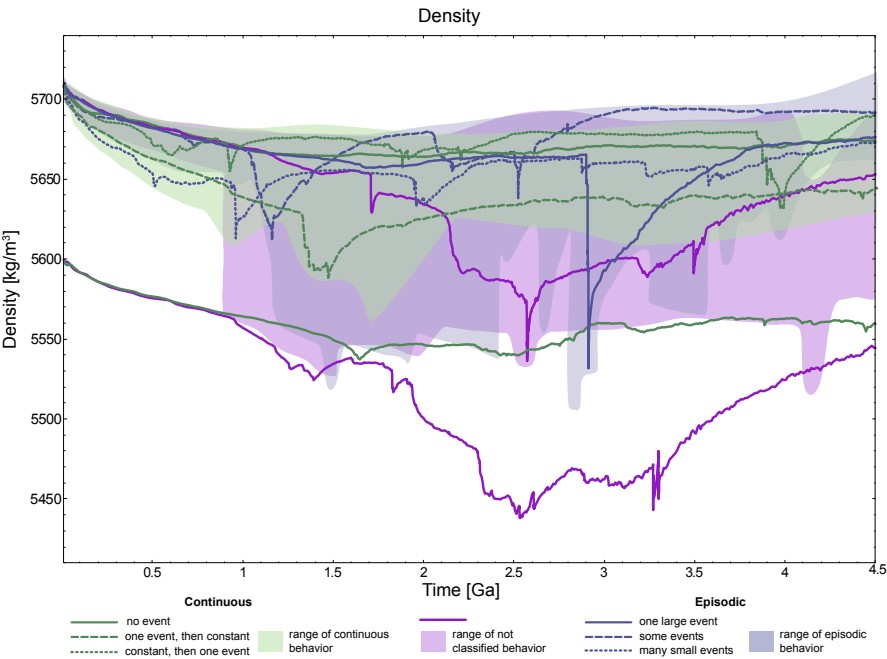

**Figure A3.** Average densities of the detected piles plotted over time. Three different behaviours are shown in different colors: continuous trend (green), episodic trend (blue) and a third, non-classified trend (purple). Sub-categories are identified by different line types. Background colors show the regions in which certain trends occur.



## A4  Pile Temperature

The average pile temperature is the only property which decreases with time. This can be attributed to the decreasing core temperature, giving as a boundary condition in our models. Anyhow, also the temperature displays either an episodic (blue) or continuous (green) behaviour (figure A4). Here, we divided both behaviours into three categories, depending on whether the temperature first increases, is more or less constant or directly decreases. Overturns lead to an increase in the average pile-temperature (figure A4, blue lines) and are followed by a strong decrease in temperature. Long episodes of stagnant lid are

responsible for the increase in or constant pile average temperature. We observe the correlations: a higher yield stress and/or a higher yield stress gradient leads to a higher average pile temperature (table 6). In contrast, a higher eruption efficiency generally results in an overall lower average pile temperature (mostly high 4200 K). High initial density of the primordial material is also related with a higher average temperature of the piles whereas a higher initial viscosity leads to the opposite. All average temperatures are in the range of 4250 K to 4450 K in the end of the simulations (figure A4).

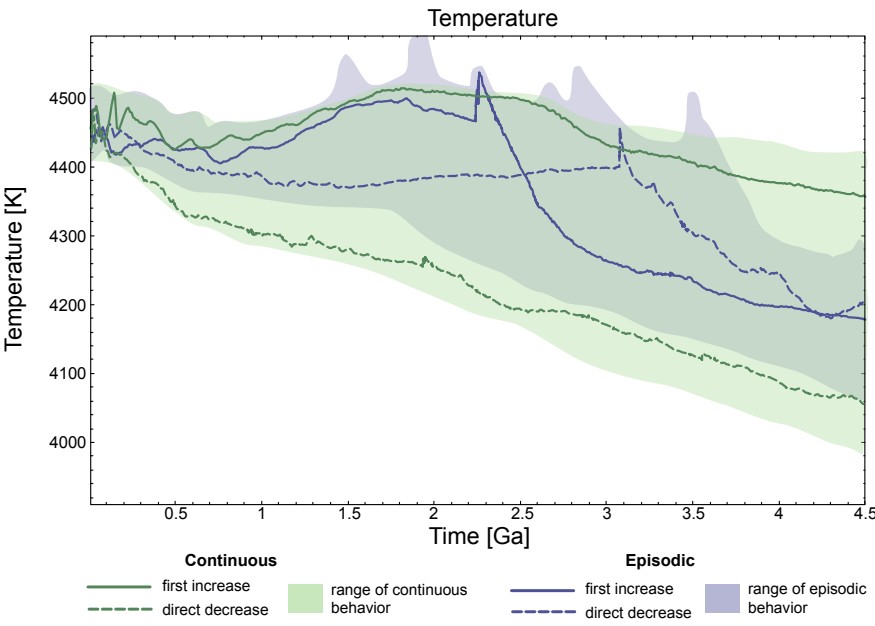

**Figure A4.** Average temperatures of the detected piles plotted over time. Two different behaviours are identified: continuous trend (green) and episodic trend (blue). Sub-categories are marked by different line types. Background colors show the regions in which certain trends occur.

*Code and data availability.*  The code is available under request.

The used, unprocessed data (for the figures) can be downloaded https://www.research-collection.ethz.ch/handle/20.500.11850/371505 and has the doi 10.3929/ethz-b-000371505.





*Author contributions.* A.R., P.T. and J.S. designed the study. P.T. developed the code. A.R. supported J.S. in setting up the model and in investigating the results. A.R. coded the grain size evolution routine and some post-processing routines. J.S. coded the pile detection and some post-processing routines, made the figures and wrote the paper draft. A.R. extended the method section and provided input and suggestions for the paper draft. P.T. gave comments and suggestions on the paper draft.

*Competing interests.* There are no competing interests present.

*Acknowledgements.* J.S. received funding from the European Union's Horizon 2020 research and innovation program under the Marie Sklodowska-Curie grant agreement 642029 - ITN CREEP, and from SNF grant 200021-182069. A.R. and P.T. were funded by ETH Zurich.




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
