# Peer review of "On the self-regulating effect of grain size evolution in mantle convection models: Application to thermo-chemical piles"

_Solid Earth, 2019_

## Referee Comment (RC1) · Bradford Foley (Referee) · 11 Dec 2019

Review of "On the self-regulating effect of grain size evolution in mantle convection models: Application to thermo-chemical piles" by Shierjott, Rozel, and Tackley

General comments:

This paper presents 2-D numerical convection models that include grain size evolution, to model the long term evolution of thermochemical piles at the base of Earth's mantle. In particular, the paper focuses on the effects of a composite rheology that includes dislocation and diffusion creep as well as a formulation for grain size evolution, to as-

sess how grain size evolution influences the dynamics of the piles. The main findings are that grain size in the piles is relatively self-regulating, following a long-term trend as a result of mantle cooling and changes in the typical stress & strain rate within the piles. Large episodic overturns lead to significant decreases in pile grain size and viscosity, but grain size quickly returns to the previous state once the overturn is over. Another important finding is that although warm temperatures in the piles lead to grain growth, this grain growth is limited by the background rate of deformational work in the piles, such that piles do not become very stiff and resistant to being pushed around the CMB by subducting slabs. I find the findings to be interesting and worthy of publication, and the science overall is sound. I do think some moderate revision is needed to more clearly highlight and demonstrate the main scientific findings, and address a few minor technical issues as well.

Specific comments:

1. This paper could be significantly improved by more clearly organizing it around central scientific questions being answered or hypotheses being tested. As of now it reads like more of a description of model results, without much direction beyond "what happens when we include grain size evolution." I have a couple suggestions for this:

A) Whether pile grain size can increase and allow the piles to become rheologically stiff, and therefore anchored at the CMB, is an interesting question, and could be looked into more thoroughly. The paper indicates that this is not the case, as the pile grain growth is limited and downwellings impacting the piles cause the piles to be rheologically weakened. This raises some questions that could be explored in more detail: What is it that prevents the piles from stiffening? Is there internal convection that supplies enough deformational work to keep grain size from growing too much? Is it downwellings hitting the piles that cause the stress/deformational work that keeps grain size from growing drastically? Likewise, during major overturns where there is significant weakening and grain size reduction of the piles, it would be useful to show the rate of deformational work in this instance

B) The fast pile grain size "recovery" is also interesting. How about using the model results to compare the recovery timescale seen from the numerical models to the theoretical prediction for recovery time, to demonstrate that the expected recover time scale indeed holds? Also, the authors should be able to work out what is stabilizing grain size and viscosity as the mantle cools (in particular for the cases shown in the appendix). There must be some trend in grain size (or viscosity) acting coupled to the change in pile temperature to keep grain size nearly constant over time. Finally, another interesting point is that grain size variations limit lateral viscosity variations; e.g. plumes have a similar viscosity to the surrounding mantle because the higher temperature is cancelled out by larger grain size. The authors could look into what conditions allow this to hold. For example, if the grain growth activation energy is much larger than the activation energy for diffusion creep, would plumes become more viscous than surrounding mantle? Or would deformation still limit the grain size?

2. Throughout this paper, the authors should be looking at the deformational work rate, not just stress. Work rate is what is controlling grain size reduction, and therefore the most relevant thing for the typical grain size in the piles and amount of grain size reduction seen when downwellings interact with the piles.

3. The authors should discuss whether the resetting of grain size at the post-perovskite phase change has any significant effect on the results, in particular for grain size evolution in the piles.

4. The results indicate diffusion creep generally dominates in the piles themselves, and dislocation creep can be active around downwellings or other high stress regions at the CMB. Given that we have observations of seismic anisotropy in some regions near the core-mantle boundary, the authors could do a more thorough comparison of their results to these observations. Comparing the settings where anisotropy is observed to where the models predict dislocation creep to be active would provide a good test to the model results.

5. Equation 7: What is the purpose of the "dislocation creep efficiency" parameter? A composite rheology formulation should be able to deal with this self-consistently, and have the temp, grain size, stress, pressure, etc dictate which mechanism dominates and controls the viscosity entirely on its own.

6. Below equation 14: "...where TCMB = 4000 K is the average temperature at the core-mantle boundary, ftop is the maximum (at 3000 K) and fbot the minimum damage fraction (at 4000 K). In order to set the damage fraction to zero at surface temperatures of 300 K, the term in (14) uses -300 in the exponent." Something's off here. By equation 14, f doesn't go to 0 at the surface, it just goes to f_top (the exponent goes to 0). Also f_top is the maximum at 300 K not 3000 K.

7. The calculation for the pile grain size recovery time for the Earth uses the typical stress and strain rate in the ambient mantle to calculate the deformational work rate. But stress and strain rate in the piles could be different. Better to analyze the flow patterns in the piles that determine the typical work rate in these regions, as I've suggested above, and use this in the estimate for the modern Earth.

Technical corrections:

Lines 42-43: I just don't follow what this sentence is trying to say

Line 101: "Intruda" likely a typo

Line 219: I think it is better to refer to this as a wattmeter since it is deformational work driving grain size reduction and not just the stress

Lines 252-253: Are the small grain sizes of 5 microns seen everywhere in the lithosphere or just at plate boundary areas?

Line 292: "This prevents the Earth to cool down more" should say prevents the Earth from cooling down more

Line 296-298: How is the second stagnant lid phase defined as stagnant lid, if surface

velocities are nearly as high as in the mobile lid phase?

Line 324: "Vigorousness" should be "vigor"

Line 406: Here is a place where the authors could look into more detail at stress and strain rate in the piles, and what sets the typical level of deformational work in the piles and hence limits grain growth

Line 480: Saying that the models can and cannot confirm the idea that plumes form at the pile edges is very confusing. If the results don't confirm this idea then they don't confirm it! Please clarify the text here.

Lines 492-493: Larger grain sizes in the plumes not affecting the viscosity: Does this mean that the viscosity is not sensitive to grain size, or that the grain size just isn't growing all that big? Confusing as written. As I suggest earlier, this issue of temperature vs. grain size tradeoffs for viscosity is something that should be looked at in more detail.

Appendix: I find this terminology of "continuous" versus "episodic" very confusing, as well as the further classification of "events, then constant," "constant, then events," etc. I'm not really sure what this classification is supposed to help the reader see. Maybe better to just show some example models individually and indicate where stagnant, mobile, and episodic overturning phases occur, so we can see how these effect the grain size evolution?

Lines 553-554: That basalt is not mixing in with the piles is an important point that needs to be explained further and compared with McNamara/Mingming Li work where they argue for basalt incorporation into piles

Appendix A3: Plotting density alone is not so useful. What really matters is the density difference between the pile and surrounding mantle. For example, the decrease in density seen due to the piles rising is not really dynamically meaningful as it is due to decompression. We need to know the density relative to surrounding mantle to see if

the buoyancy has changed.

---

## Referee Comment (RC2) · Anonymous Referee #2 · 7 Jan 2020

This manuscript presents the results of 2D simulations in spherical geometry investigating the effect of grain size evolution, with application to thermo-chemical piles. Grain size is important because it affects viscosity, and it modifies the effective temperature-dependence of the viscosity. Modeling grain size is challenging, because grain size depends in a complex way on several parameters, such as stresses, phase transitions, temperature and composition. The manuscript is worth publication, after revision.

The manuscript reads as a diligent description of model results, but, all in all, it seems a bit pedantic. In particular, the connection between lithospheric processes and the deep-seated thermo-chemical piles remains unclear. For example, in line 337 we read

that "the pile-temperature mostly depends on the eruption efficiency", but it is never explained how the eruption efficiency (i.e., the percentage of basalts erupted at the surface or intruded as gabbros) can effect the temperature of thermo-chemical piles at the base of the Earth's mantle. Moreover, the reader never understands the internal dynamics of the pile (velocity field, internal convection, mixing with subducted material ecc). It is also impossible to understand how the authors obtain (50%!) melting at the base of the mantle, nor how melting would affect viscosity or grain size. In other words, if the focus of the manuscript are the piles, then the authors should be more specific and quantitative.

The novelty of the simulations resides in the composite rheology and in the fact that viscosity is grain size-dependent. This aspect should be presented more clearly, already in the introduction, where the reader expects to find a pedagogic and insightful presentation of diffusion and dislocation creep (you do it in paragraph 2.3, lines 155, but I think it comes too late). The paragraph you have in the introduction (starting at line 68) is too technical (for example your sentence "grain growth when conditions favor high grain boundary energy" needs to be better explained). I also suggest to expand the few lines describing diffusion-dislocations creep in the mantle (for example, your sentence "However, several other studies indicate that in many regions dislocation creep is active" is too dry and we do not learn much, nor do we gain insight to compare previous studies to your new results). In the introduction we should also talk about seismic anisotropy.

In the following I give my comments (in a line by line order). Line 31: it is the opposite!! Pacific LLSVP is roundish. African LLSVP is elongated. Lines 40-44: it would improve by being more specific (i.e., quantify density differences, and how they vary with depth). Line 50: I would add a citation : U. Christensen, A.W. Hofmann, Segregation of subducted oceanic crust in the convecting mantle, J. Geophys. Res. 99 (1994) 19867-19884. Line 53: I find this sentence useless ("Since LLSVPs remain physically unreachable numerical and experimental studies try to constrain the parameter space"). Line 62: Here you should say more, and your sentence " Only very few studies have considered a composite and grain size-dependent viscosity [ref]" is unsatisfactory. At this point the reader needs to understand: (1) what previous authors have done and found, (2) what is new in your work with respect to what has been already published. Line 74: Your sentence " Among others, Cordier et al. (2004) suggested...." skips to cite previous papers before Cordier et al. (2004). I do not recommend this practice. Line 82: Your sentence "By also considering a primordial layer we are able to elaborate on the origin of LLSVPs" (e.g. subducted basalt, primordial reservoir or the basal mélange (Tackley, 2012))" does not seem true to me, since (1) you are never specific about the composition and internal dynamics of the pile, (2) you do not span a range of buoyancy ratio B. Line 85: Your sentence " We investigate whether piles behave as obstacles to convection, whether they get pushed around or even entrained by mantle flow" also does not seem true to me, since you never quantify entrainment, you only say that they are pushed around, but this is well known. Line 98: Your sentence "If the melt is generated at a depth lower or equal to 300km, the basalt is..." seems incorrect. If partial melting occurs at 300km depth the liquid composition cannot possibly be a basalt (already at 100km depth the melt has a picritic composition). I suggest to add a citation to strengthen your statement. Line 101: Your sentence "Intruda is therefore warmer than the ambient lithosphere which results in lithosphere-weakening" needs to be explained, namely for the "lithosphere-weakening" part. What are the modeled melting rates? Over which length-scales do you intrude the lithosphere? Over which time-scales do the intrusions cool? How correctly can you solve for lithospheric processes knowing that your grid resolution is quite poor (512 elements for 360° means that at lithospheric depths your element size is 78km). Do you consider latent heat of melting? Line 120: Your definition of the Buoyancy number is confusing/wrong: (1) The numerator is a density difference (RHOprimordial - RHOsurrounding mantle). Why do you use (RHOprimordial - RHObasalt), your mantle is NOT a basalt, it is 80% harzburgite and 20% basalt. (2) The denominator is also problematic, since RHO0 is NOT the average of RHOprimordial and RHObasalt, but it must be the RHO entering in the

[Figure]

Rayleigh number (never given in the tables). Line 124-125: The simple statement "we vary the intensity of dynamic recrystallization" needs to be explained. What is the physics behind? What does this mean? Line 141: Rewrite eq.(2) Line 148: Rewrite the last term of eq. (4). In the equation you have an internal heating term, but we never find the value of H. Line 201: The definition of full mechanical work is wired, I guess a typing problem. (Check also eq. 15 and 16). Line 207: Here we find Tcmb=4000K, whereas in Table 1 Tcmb=5000K. Why? More generally, your ftop, fbot, and the physics behind eq. (14) are unclear. Line 220: Your criteria to detect the pile (>90% of primordial + basalt) makes it impossible to detect entrainment of surrounding mantle into the pile. For example, if you have (80% primordial + 20% surrounding mantle) how is this considered? Normal mantle ? Line 225: I do not understand eq. 18: Tpile > (3000K + Tcmb). In table 1 Tcmb=5000K, so how can Tpile be greater than (3000+Tcmb)? It means Tpile=8000K, which is impossible. Line 244 and Table 3, Table 4: Warning !! In the text (line 244) we read that for density 3140 kg/m3 the ratio B=0.14, BUT, reading the figure caption of Table 3 we find that when the density is 3140 kg/m3 the ratio B=0.24. Line 247 and elsewhere: The ratio of strain rate due to dislocation creep and strain rate due to diffusion creep is defined as "rheology" and throughout the rest of the manuscript "rheology" has this meaning. I think this is very confusing and I invite you to call it "the rheology ratio" but not "the rheology". Line 248 and Figure 3: Warning, two panels are never mentioned, neither in the text nor in the figure caption. I'm talking about the two panels at the left. What are the green lines? (In line 287 you say something about figure 3, while presenting figure 4, and also in line 461.... well, all this is poorly organized). Line 250 and Figure 3: (1) it is very hard to detect the white and the black lines. (2) partial melt higher than 50% ! This is very high, but in the text you never talk about partial melting, you never provide the solidus used..... (3) In the lower mantle it is incorrect to talk about basalt, you should use "basaltic composition". (4) It 's impossible to see that "basalt is pushed aside". You need to have a figure with a zoom on the region of interest. Line 261: I do not understand what do you mean by "the newly formed parts of the pile". Since the pile does not entrain, but it is merely displaced,

how do you generate "newly formed parts of the pile" ?? Line 268: Provide the solidus used to calculate melting at the CMB. Line 270: It is wired: first we read that "basaltic material " melts up to 50%, and then we read "once the basaltic material has warmed up". How is this possible? Your statements are neither quantified nor justified. Show a P-T diagram with real temperatures and the used solidus for each composition, and then the reader will understand. Line 293, line 295, and Figure 4: I do not understand why the pile density varies. Line 301 and Figure 4: The modeled surface velocities can be higher than 10e3 cm/yr and up to 10e4 cm/yr, these values are huge (10-100 m/yr!!) and deserve a comment. Only saying "a lot of cold lithosphere simultaneously moves down" is insufficient. You need to quantify subducted volumes and you need to convince the reader that surface velocities at 10m/yr are not an artifact of the numerical simulation. Line 310: I do not understand why the density of the pile changes because of "relocation" of pile material". Density variations caused by pressure variations are not an intrinsic density change, they are just an effect of compression/decompression. Lines 439 to 443: Rewrite. Line 470: Provide reference of articles suggesting that piles "spatially determine subduction zones". Line 509: Why is pile density self-regulating ??

Final comment: once you have reviewed the manuscript I suggest to rewrite parts of the abstract in a more concise, punchy, way.

---

## Author Comment (AC1) · 4 Feb 2020

**1 Response to Reviewer #1: Bradford Foley**

Review of "On the self-regulating effect of grain size evolution in mantle convection models: Application to thermo-chemical piles" by Schierjott, Rozel, and Tackley

**General comments:**

This paper presents 2-D numerical convection models that include grain size

evolution, to model the long term evolution of thermochemical piles at the base of Earth's mantle. In particular, the paper focuses on the effects of a composite rheology that includes dislocation and diffusion creep as well as a formulation for grain size evolution, to assess how grain size evolution influences the dynamics of the piles. The main findings are that grain size in the piles is relatively self-regulating, following a long-term trend as a result of mantle cooling and changes in the typical stress strain rate within the piles. Large episodic overturns lead to significant decreases in pile grain size and viscosity, but grain size quickly returns to the previous state once the overturn is over. Another important finding is that although warm temperatures in the piles lead to grain growth, this grain growth is limited by the background rate of deformational work in the piles, such that piles do not become very stiff and resistant to being pushed around the CMB by subducting slabs. I find the findings to be interesting and worthy of publication, and the science overall is sound. I do think some moderate revision is needed to more clearly highlight and demonstrate the main scientific findings, and address a few minor technical issues as well.

**Specific comments:**

1. This paper could be significantly improved by more clearly organizing it around central scientific questions being answered or hypotheses being tested. As of now it reads like more of a description of model results, without much direction beyond "what happens when we include grain size evolution." I have a couple suggestions for this: A) Whether pile grain size can increase and allow the piles to become rheologically stiff, and therefore anchored at the CMB, is an interesting question, and could be looked into more thoroughly. The paper indicates that this is not the case, as the pile grain growth is limited and downwellings impacting the piles cause the piles to be rheologically weakened. This raises some questions that could be explored in more detail: What is it that prevents the piles from stiffening? Is there internal convection that supplies enough deformational work to keep grain size from growing too much?
Is it downwellings hitting the piles that cause the stress/deformational work that keeps grain size from growing drastically? Likewise, during major overturns where there is significant weakening and grain size reduction of the piles, it would be useful to show the rate of deformational work in this instance.

¶Indeed we had a hard time deciding how to present the results of our study. We first formulated several scientific questions but obtained a very complicated structure with redundancies. In the end we decided to first offer a global presentation of the fields, followed by 0D averages for each convection regime, 1D profiles ordered by convection regimes, and only then attempt to answer scientific questions.

¶Thus, we do not think that, at this point, changing the structure of the paper through minor revisions would help in clarifying scientific questions. However, we do answer your points in the Discussion section, where clarifications fit well into the design of the paper.

¶In short, we answer the following scientific questions (also in the paper):

- Ambient mantle mechanical conditions (stress and strain rate) reach and propagate through the thermo-mechanical piles. In other words, we find that the piles are not mechanically decoupled from the mantle. Therefore, the idea that the piles can be much stronger than the mantle is not supported by our results. This regime might exist, we just did not observe it in our simulation using experimental (reasonable) coefficients. Moreover, this means that the viscosity of the piles does change with the convection regime as stress and strain rates vary.
- Yes convection stresses keep the grains from growing too large. This is shown in figure 3 (equilibrium grain size vs time), Eq. 16 and discussed in section 3.4. More precisely, mechanical work, as you say, controls the grain size.
- both downwelling and upwellings generally contribute to the ambient mechanical work. It would be very hard to know exactly if downwellings or upwellings dominate the ambient mechanical conditions but we observe that downwellings are

SED
important. This can be explained by the fact that the bottom boundary layer is not potentially unstable like the lithosphere is in the episodic regime. Solomatov (2004) does attempt to answer the question of partitioning of stress contributions between upwellings and downwellings (as you know), but his study was performed in a very simplified framework, which might not fully apply in our case.

 Unfortunately, at this stage, we cannot plot the mechanical work itself without quite some programming (or rerunning all cases). However, one can have an idea of what the mechanical work would be by multiplying the stress and strain rate invariants. Figure 3 shows that both those fields are relatively homogeneous around large structures (either whole mantle or around a large downwelling during an overturn) so the mechanical work is very likely to also be rather homogeneous.

B) The fast pile grain size "recovery" is also interesting. How about using the model results to compare the recovery timescale seen from the numerical models to the theoretical prediction for recovery time, to demonstrate that the expected recover time scale indeed holds? Also, the authors should be able to work out what is stabilizing grain size and viscosity as the mantle cools (in particular for the cases shown in the appendix). There must be some trend in grain size (or viscosity) acting coupled to the change in pile temperature to keep grain size nearly constant over time. Finally, another interesting point is that grain size variations limit lateral viscosity variations; e.g. plumes have a similar viscosity to the surrounding mantle because the higher temperature is cancelled out by larger grain size. The authors could look into what conditions allow this to hold. For example, if the grain growth activation energy is much larger than the activation energy for diffusion creep, would plumes become more viscous than surrounding mantle? Or would deformation still limit the grain size?

¶These questions are indeed very important from a fundamental point of view. Some of them are answered in another article in preparation, which should have been

**SED**
published before the present manuscript but technical difficulties made it impossible to finish as it explores a much larger parameter space and answers theoretical questions. Still we can partially answer your requests:

- The recovery time scale is a very parameter-dependent quantity. We chose to mention its existence in our discussion but we do not to claim that all parameters leading to its estimation are known in a robust way. We rather give an estimation and do not attempt more. We think the idea that stresses penetrating through piles might hold for a large range of rheological and mineralogical parameters but the grain size itself in the pile is hard to really assess. Since the petrological nature of the LLSVPs is highly uncertain, we chose not to provide a prediction, only an estimation.
- Yes we did want to mention the competition between temperature and grain size. We have a dedicated paragraph on this topic (section 3.4). The paper in preparation will be able to answer more on this idea that the difference of activation energies of growth and rheology will dominate (and even potentially invert) the temperature-dependence of the rheology. Since this idea has been proposed in the past (Solomatov and Korenaga do mention this) we did not detail it too much in the present paper. Overall, still our observation that stress does propagate through the LLSVPs seems to indicate that stresses would also make it through viscous plumes. We observe that mechanical quantities tend to homogenise in the mantle and through whichever anomaly.

2. Throughout this paper, the authors should be looking at the deformational work rate, not just stress. Work rate is what is controlling grain size reduction, and therefore the most relevant thing for the typical grain size in the piles and amount of grain size reduction seen when downwellings interact with the piles.
¶We added to Figure 4 a plot of the average work rate occurring in the pile (replacing the plot of average density). From this plot we can see that when stress is high the work rate is also high. Hence, our interpretation does not change. In any case, we agree, the work rate is better and now our paper has a much stronger argument than before.

3. The authors should discuss whether the resetting of grain size at the postperovskite phase change has any significant effect on the results, in particular for grain size evolution in the piles.

¶The influence of the post-perovskite phase change is negligible because grains grow back very fast in any case due to a low deformational work rate and high temperatures close to the CMB. ¶Moreover, the radial velocities are usually small so a very limited volume of material goes through the Post-Perovskite phase transition. We have added comments on this in the text.

4. The results indicate diffusion creep generally dominates in the piles themselves, and dislocation creep can be active around downwellings or other high stress regions at the CMB. Given that we have observations of seismic anisotropy in some regions near the core-mantle boundary, the authors could do a more thorough comparison of their results to these observations. Comparing the settings where anisotropy is observed to where the models predict dislocation creep to be active would provide a good test to the model results.

¶We have edited the paragraph and added some details:

The anisotropy observed in some parts of the D"-layer (Lay and Young, 1991; Lay et al., 1998; Garnero, 2000; Kendall and Silver, 1996), specifically in regions of high stress (Karato, 1998), can be explained by regionally occurring dislocation creep due to downwelling-induced high stresses as has been proposed by (Karato, 1998). Seismic anisotropy resulting from dislocation creep in the rest of the D"-layer can better
be explained by material layering, aligned inclusions or flow fabrics due to a strongly sheared thermal boundary layer and crystalline alignment as has been suggested by for example Kendall and Silver (1996) and Doornbos et al. (1986), respectively.

5. Equation 7: What is the purpose of the "dislocation creep efficiency" parameter? A composite rheology formulation should be able to deal with this self-consistently, and have the temp, grain size, stress, pressure, etc dictate which mechanism dominates and controls the viscosity entirely on its own.

¶Sorry, we have reformulated the text to explain this better. The rheological coefficients used in  $\eta_{df}$  and  $\eta_{ds}$  would independently lead to the viscosity profile of the Earth for both diffusion and dislocation creep if the global stress and strain rate of the Earth occurred (e.g., in case of plate tectonic). So if we solely used diffusion creep or solely dislocation creep, we would probably obtain the viscosity profile of the Earth. However, this is not what we want here. We rather want to have diffusion creep dominating in the lower mantle and dislocation creep dominating in the upper mantle. The dislocation creep efficiency is a number we have defined to favour diffusion or dislocation independently in the upper and lower mantle. This does not mean that the rheology is forced at all times. The rheology (effective dislocation creep/diffusion creep fraction) still depends on stress, grain size, pressure, etc., is time-dependent and depends on the self regulating processes happening during convection. But if plate tectonics occurs, then the effective rheology will be the one predicted by the dislocation creep efficiency.

6. Below equation 14: ": : :where TCMB = 4000 K is the average temperature at the core-mantle boundary, ftop is the maximum (at 3000 K) and fbot the minimum damage fraction (at 4000 K). In order to set the damage fraction to zero at surface temperatures of 300 K, the term in (14) uses -300 in the exponent." Something's off here. By equation 14, f doesn't go to 0 at the surface, it just goes to ftop (the exponent
goes to 0). Also ftop is the maximum at 300 K not 3000 K.

¶Yes indeed the text was wrong. The equation is correct. We have changed the text to:

where  $T_{\text{CMB}}$  = 4000 K is the average temperature at the core-mantle boundary,  $f_{\text{top}}$  is the maximum (at 300 K), and  $f_{\text{bot}}$  the minimum damage fraction (at 4000 K).

7. The calculation for the pile grain size recovery time for the Earth uses the typical stress and strain rate in the ambient mantle to calculate the deformational work rate. But stress and strain rate in the piles could be different. Better to analyze the flow patterns in the piles that determine the typical work rate in these regions, as I've suggested above, and use this in the estimate for the modern Earth.

¶If one thinks that stress and strain rate are different inside and outside the piles, then indeed using global mantle flow kinetics to estimate pile conditions would not be meaningful concerning the piles. However, our plots of the 1D profiles inside and outside the pile indicate that the viscosity is similar in the pile and in the surrounding mantle. In such case, the ambient flow should be a good indication of the pile conditions.

¶We were first aiming at an article in which numerical simulations would be carefully compared to Earth observations. However since grain size evolution makes it hard to obtain the mobile-lid regime, we did not obtain a large set of simulations with a behavior comparable to that of the Earth. Nevertheless, we were surprised about the self-regulating behavior of the pile for each convection regime so we decided to write the present paper. However, we do not believe our study is general enough to make an actual comparison with the Earth, we would rather simply provide estimates.
**Technical corrections:**

Lines 42-43: I just don't follow what this sentence is trying to say ¶We have changed the sentence to "By analysing deep mantle-sensitive Stoneley mode data in a joint P- and S-wave inversion this recent work showed that at least the upper parts of LLSVPs might be lighter than the ambient mantle."

Line 101: "Intruda" likely a typo ¶We changed it to "Intruded material is

Line 219: I think it is better to refer to this as a wattmeter since it is deformational work driving grain size reduction and not just the stress ¶We removed piezometer.

Lines 252-253: Are the small grain sizes of 5 microns seen everywhere in the lithosphere or just at plate boundary areas?

¶They are mainly that small in areas of plate boundaries. In the rest of the lithosphere they can be large as 100  $\mu$ m. We added "Small grains (around 5  $\mu$ m in plate boundary areas and up to 100  $\mu$ m elsewhere)....".

Line 292: "This prevents the Earth to cool down more" should say prevents the Earth from cooling down more ¶We have changed the wording to the suggested phrase.

Line 296-298: How is the second stagnant lid phase defined as stagnant lid, if surface velocities are nearly as high as in the mobile lid phase?

¶The stagnant lid phase is defined to be when the average surface velocity is less than 1cm/yr. Although the surface velocity is close to this threshold in the second stagnant lid phase, the simulations don't show rapid overturns or subduction events so it can be classified as stagnant lid. After 4.3 Gyr there is some mobile component. We
distinguish this now in the text:

"During the second stagnant lid phase (3.5-4.3 Gyr) .... [...] The pile temperature can further decrease during the second stagnant lid phase because there still exists some movement at the surface, manifested by dripping of lithosphere."

Line 324: "Vigorousness" should be "vigor" ¶changed to vigor

Line 406: Here is a place where the authors could look into more detail at stress and strain rate in the piles, and what sets the typical level of deformational work in the piles and hence limits grain growth

¶We now plot the mechanical work rate in the pile as a function of time.

Line 480: Saying that the models can and cannot confirm the idea that plumes form at the pile edges is very confusing. If the results don't confirm this idea then they don't confirm it! Please clarify the text here.

¶We have edited the paragraph to:

Our thermo-chemical piles are also not surrounded by plume generation zones (PGZ), as suggested by Burke et al. (2008), but plumes rise directly from the piles as well as from their margins. They, as others (Torsvik et al. (2006), Torsvik et al. (2010)), conclude that LLVPs (in geodynamics referred to as thermo-chemical piles) have been stable in time because the downward projection of Large Igneous Province (LIP) sites can be linked to the margins of LLSVPs after rotating them back to their original eruption sites. LIPs in the 200 to 500 Myr age range let them conclude that LLSVPs have been occupying the same location for the same duration. Stable piles can only be confirmed with our models in the case of the absence of strong downwellings (subduction zones), hence for the last 200 to 500 Myr because we observe that downwellings govern the piles' spatial distribution. If there are no strong downwelling
events disturbing the location of the piles, we can observe piles stable for at least 300 Myr. However, without dominant downwellings, we do not see plate tectonic-like behaviour in our simulations, implying that we either observe stable piles or plate tectonic-like behaviour but not both simultaneously. Even without a plate tectonic-like convection regime in our models, it is difficult to draw conclusions about the actual stability and spatial distribution of LLSVPs. Problematic is that we neither employ realistic plate velocities, nor use three-dimensional models.

Lines 492-493: Larger grain sizes in the plumes not affecting the viscosity: Does this mean that the viscosity is not sensitive to grain size, or that the grain size just isn't growing all that big? Confusing as written. As I suggest earlier, this issue of temperature vs. grain size tradeoffs for viscosity is something that should be looked at in more detail.

¶We have edited this part to:

"Our results show that grain size has a great impact on the viscosity in numerical convection models. Similar to results by Dannberg et al. (2017), we observe strong lateral variations in grain size and resulting viscosity in our simulations, particularly during resurfacings or prominent downwellings. Overturn events lead to a distinct 'bimodal' behaviour in which one half of the spherical annulus shows a distinct decrease in viscosity and smaller grain size than the other half (figure 3, 1.58 Gyr). Downgoing slabs are surrounded by regions with lower grain size, high strain rate and reduced viscosity. This finding agrees well with what Dannberg et al., (2017) reported. However, in times without any particular downwelling event we do not observe strong lateral viscosity variations in the lower mantle. Viscosity is relatively uniform having values between  $5 \times 10^{22}$  Pa s (around piles and regions of high melt content) and  $5 \times 10^{24}$  Pa s (regions with high melt content).

Most of the lower mantle has a viscosity on the order of  $5 \times 10^{23}$  Pa s. Solomatov Moresi (1996), Karato Rubie (1997), Solomatov et al. (2002) and Korenaga (2005) suggest that higher temperatures in plumes could result in higher viscosity due to

SED
larger grains. This suggestion cannot be supported with our simulations, but might be probable if different grain growth parameters, for example stronger grain growth, were used. In our simulations, the expected increase in viscosity due to larger grain size in plumes is buffered by the higher temperature of the plume itself. The surprisingly high viscosity of regions with a high melt fraction is not a physical observation but results from how the overall viscosity is computed. We only use the grain size in the solid matrix to compute the viscosity and neglect the impact of the melt content which is usually fine, which is usually fine except for regions with a particularly high melt content."

**Appendix:**

I find this terminology of "continuous" versus "episodic" very confusing, as well as the further classification of "events, then constant," "constant, then events," etc. I'm not really sure what this classification is supposed to help the reader see. Maybe better to just show some example models individually and indicate where stagnant, mobile, and episodic overturning phases occur, so we can see how these effect the grain size evolution?

¶Generally, the results all show the same behavior, meaning we see large drops in grain size right after an overturn event, or a relatively constant grain size if the run does not show any overturn or downwelling events. The appendix arose from the fact that we initially decided to structure the paper differently, where we tried to find dependencies of the constant or episodic behaviour on the input parameters. However, this proved to be impossible and we re-structured the paper around the stagnant lid, plate-tectonic-like and overturn phase. The figures in the end are only there to demonstrate that the simulation results of the pile material show a similar behavior and basically only depend on the convection regime. We have removed the appendix since the figures don't really help to understand the points we try to make in the paper.
Lines 553-554: That basalt is not mixing in with the piles is an important point that needs to be explained further and compared with McNamara/Mingming Li work where they argue for basalt incorporation into piles

¶This part we have removed. We realise that it is interesting and might be of high importance but we didn't study this observation in detail, therefore we cannot give any detailed results or explanation.

Appendix A3: Plotting density alone is not so useful. What really matters is the density difference between the pile and surrounding mantle. For example, the decrease in density seen due to the piles rising is not really dynamically meaningful as it is due to decompression. We need to know the density relative to surrounding mantle to see if the buoyancy has changed.

¶We have removed the appendix. We decided, following the comments, that the appendix does not add anything valuable to the paper.

---

## Author Comment (AC2) · 4 Feb 2020

**1  Response to Reviewer #2: Anonymous reviewer**

This manuscript presents the results of 2D simulations in spherical geometry investigating the effect of grain size evolution, with application to thermo-chemical piles. Grain size is important because it affects viscosity, and it modifies the effective temperature dependence of the viscosity. Modeling grain size is challenging, because grain size depends in a complex way on several parameters, such as stresses, phase transitions, temperature and composition. The manuscript is worth publication, after

revision. The manuscript reads as a diligent description of model results, but, all in all, it seems a bit pedantic.

In particular, the connection between lithospheric processes and the deep-seated thermo-chemical piles remains unclear. For example, in line 337 we read that "the pile-temperature mostly depends on the eruption efficiency", but it is never explained how the eruption efficiency (i.e., the percentage of basalts erupted at the surface or intruded as gabbros) can effect the temperature of thermo-chemical piles at the base of the Earth's mantle.

We have added an explanation on this to the manuscript. In fact this is surprisingly simple: when most of the melt is erupted, the lithosphere is thick and therefore cools the LLSVPs very well when it reaches the CMB. When most of the melt is intruded, the lithosphere is thin and tends to drip down instead of exhibiting large-scale resurfacing events. Anyhow, the focus of the paper is not the link between LLSVPs and lithospheric processes. The change of eruption efficiency only arose because we aimed for Earth-like convection regimes and eruption efficiency is a potential way to receive it in whole-Earth geodynamic models.

Moreover, the reader never understands the internal dynamics of the pile (velocity field, internal convection, mixing with subducted material ecc).

Indeed this is a disappointing problem also for us. We cannot really see the internal dynamics of LLSVPs as we are computing long term mantle dynamics. The resolution is rather low so we chose to only look into pile averages to try to report a result as robust as possible. Increasing the resolution is very difficult as the simulations already took a very long time to run. We re-wrote the focus and goals of the paper slightly to make it easier to grasp that not the internal behavior of the piles is the focus but their interaction with the mantle and their properties. We didn't observe any mixing, wherefore we did not focus our paper around this topic. It may have been worth to investigate further in which cases mixing would have been observed, but this would have meant a different scope of the paper and a different parameter study. Although

we state in a few sentence that there is no mixing, we will remove it because we do not provide a detailed study on this topic.

It is also impossible to understand how the authors obtain (50%!) melting at the base of the mantle, nor how melting would affect viscosity or grain size.
We now provide much more information on melting and crust production in the text.
In other words, if the focus of the manuscript are the piles, then the authors should be more specific and quantitative.
We add statements in the article to explain that we are looking at the big picture instead of details as we run long term simulations with limited resolution. We try to state more clearly now that the goal of the paper is to demonstrate the general behavior and evolution of LLSVPs and their influence on the overall dynamics of the Earth's mantle instead of detailed internal convection or small-scale mixing. We try to provide numbers and be quantitative, but specific numbers are difficult to provide since grain size evolution parameters themselves are highly uncertain. Therefore, we provide averages of pile properties which is already more advanced and quantitative than other 'pile-paper'.

The novelty of the simulations resides in the composite rheology and in the fact that viscosity is grain size-dependent. This aspect should be presented more clearly, already in the introduction, where the reader expects to find a pedagogic and insightful presentation of diffusion and dislocation creep (you do it in paragraph 2.3, lines 155, but I think it comes too late). The paragraph you have in the introduction (starting at line 68) is too technical (for example your sentence "grain growth when conditions favor high grain boundary energy" needs to be better explained). I also suggest to expand the few lines describing diffusion-dislocations creep in the mantle (for example, your sentence "However, several other studies indicate that in many regions dislocation creep is active" is too dry and we do not learn much, nor do we gain insight to compare previous studies to your new results). In the introduction we should also talk about

seismic anisotropy.

We have added a paragraph describing previous whole-Earth studies that use grain size in some sort or another in their rheology definition.

**In the following I give my comments (in a line by line order).**

Line 31: it is the opposite!! Pacific LLSVP is roundish. African LLSVP is elongated.

Yes. This was a typo that we have now corrected.

Lines 40-44: it would improve by being more specific (i.e., quantify density differences, and how they vary with depth).

We have added the estimated density difference in the text.

Line 50: I would add a citation: U. Christensen, A.W. Hofmann, Segregation of subducted oceanic crust in the convecting mantle, J. Geophys. Res. 99 (1994) 19867-19884.

We have added it.

Line 53: I find this sentence useless ("Since LLSVPs remain physically unreachable numerical and experimental studies try to constrain the parameter space").

We have removed it.

Line 62: Here you should say more, and your sentence " Only very few studies have considered a composite and grain size-dependent viscosity [ref]" is unsatisfactory.

At this point the reader needs to understand: (1) what previous authors have done and found, (2) what is new in your work with respect to what has been already published.

Yes. We have add much more detail on this, as suggested.

[Figure]

Line 74: Your sentence " Among others, Cordier et al. (2004) suggested...." skips to cite previous papers before Cordier et al. (2004). I do not recommend this practice.
We have added the earliest citation (to our knowledge). There are not that many actually.

Line 82: Your sentence "By also considering a primordial layer we are able to elaborate on the origin of LLSVPs" (e.g. subducted basalt, primordial reservoir or the basal mélange (Tackley, 2012))" does not seem true to me, since (1) you are never specific about the composition and internal dynamics of the pile, (2) you do not span a range of buoyancy ratio B.
True, we indeed did not mention the mixing of basalt and pile material and entrainment of pile material in the ambient mantle a lot. We have therefore removed this and stated the other points of our paper more clearly, as you suggest below.

Line 85: Your sentence " We investigate whether piles behave as obstacles to convection, whether they get pushed around or even entrained by mantle flow" also does not seem true to me, since you never quantify entrainment, you only say that they are pushed around, but this is well known.
We observe that stresses and strain rates propagate through the pile, therefore they are certainly involved in the global deformation. We also actually see the piles moving with the flow. Indeed we do not quantify entrainment but the fact the piles are pushed around is clear in our results. Actually a lot of people believe that piles are fixed, following what the people from CEED are claiming. This is the reason why we have written this statement here.

Line 98: Your sentence "If the melt is generated at a depth lower or equal to 300 km, the basalt is..." seems incorrect. If partial melting occurs at 300 km depth the

liquid composition cannot possibly be a basalt (already at 100 km depth the melt has a picritic composition). I suggest to add a citation to strengthen your statement.
Apologies, this is one of our common mistake, we mistake eclogitic melt for basaltic melt as the eclogite becomes basalt at the surface in our code. We correct this in the manuscript.

Line 101: Your sentence "Intruda is therefore warmer than the ambient lithosphere which results in lithosphere-weakening" needs to be explained, namely for the "lithosphere-weakening" part. What are the modeled melting rates? Over which length-scales do you intrude the lithosphere? Over which time-scales do the intrusions cool? How correctly can you solve for lithospheric processes knowing that your grid resolution is quite poor (512 elements for 360 degrees means that at lithospheric depths your element size is 78km). Do you consider latent heat of melting?
We did not detail this much as this paper is not focusing on melting and crust production. All of these questions are answered in a manuscript of Diogo Lourenço (and A. Rozel and P. Tackley) that is still in its last round of review (now minor revisions), and also in the doctoral thesis of Diogo Lourenço (online on ETH's web site). We answer your questions by adding clarifications in the text.

Line 120: Your definition of the Buoyancy number is confusing/wrong:
(1) The numerator is a density difference (RHOprimordial - RHOsurrounding mantle). Why do you use (RHOprimordial - RHObasalt), your mantle is NOT a basalt, it is 80% harzburgite and 20% basalt.
(2) The denominator is also problematic, since RHO0 is NOT the average of RHOprimordial and RHObasalt, but it must be the RHO entering in the Rayleigh number (never given in the tables).
We have removed this part because the buoyancy number in any case does not play a significant role in our paper since we do not investigate a vast parameter space of the primordial material.

Line 124-125: The simple statement " we vary the intensity of dynamic recrystallisation" needs to be explained. What is the physics behind? What does this mean?
We have clarified this in the text.

Line 141: Rewrite eq.(2)
We have corrected this typo.

Line 148: Rewrite the last term of eq. (4). In the equation you have an internal heating term, but we never find the value of H. Line 201: The definition of full mechanical work is wired, I guess a typing problem. (Check also eq. 15 and 16).
We now detail in the manuscript what tensor contraction is (this is a bit unusual indeed). We have also added the original radiogenic power $H_0$, the decay half life and the partitioning coefficient of heat sources during melting in table 1.

Line 207: Here we find Tcmb=4000K, whereas in Table 1 Tcmb=5000K. Why? More generally, your ftop, fbot, and the physics behind eq. (14) are unclear.
We now distinguish the values mentioned in table 1 and in the text.

Line 220: Your criteria to detect the pile (>90% of primordial + basalt) makes it impossible to detect entrainment of surrounding mantle into the pile. For example, if you have (80% primordial + 20% surrounding mantle) how is this considered? Normal mantle ?
We define that the piles' composition must be primordial material but can include some basaltic composition and even up to 10% of ambient mantle. Every cell that contains some percentage of primordial material is considered pile for sure. But it can also only contain 30% of primordial material and 60% of basalt and 10% of ambient mantle and will still be considered pile.

Line 225: I do not understand eq. 18: Tpile > (3000K + Tcmb). In table 1 Tcmb=5000K, so how can Tpile be greater than (3000+Tcmb)? It means Tpile=8000K, which is impossible.
Sorry, this is a mistake. It is meant to be divided by 2.

Line 244 and Table 3, Table 4: Warning !! In the text (line 244) we read that for density 3140 kg/m3 the ratio B=0.14, BUT, reading the figure caption of Table 3 we find that when the density is 3140 kg/m3 the ratio B=0.24.
We have removed the buoyancy ratio from the manuscript.

Line 247 and elsewhere: The ratio of strain rate due to dislocation creep and strain rate due to diffusion creep is defined as "rheology" and throughout the rest of the manuscript "rheology" has this meaning. I think this is very confusing and I invite you to call it "the rheology ratio" but not "the rheology".
Sorry, but we would rather not change this expression. Since we define "rheology" in the beginning and on every figure it should be clear to the reader. Furthermore, since rheology is the study of deformation of material we do not see a problem with calling the dominant deformation mechanism "rheology". We also always explicitly state which deformation is dominant.

Line 248 and Figure 3: Warning, two panels are never mentioned, neither in the text nor in the figure caption. I'm talking about the two panels at the left. What are the green lines? (In line 287 you say something about figure 3, while presenting figure 4, and also in line 461.... well, all this is poorly organized).
we now mention the panels explicitly. The organisation follows the convection regimes. Several figures illustrate different aspects of each convection regime. This is the reason why we cite 2 figures together.

Line 250 and Figure 3:

(1) it is very hard to detect the white and the black lines.

We first plotted these figures with thicker lines but it becomes harder to see the pile fields (too strongly overlapped with the lines).

(2) partial melt higher than 50% ! This is very high, but in the text you never talk about partial melting, you never provide the solidus used.....

We now provide the solidus temperature function. Yes indeed, high melt fractions can seem unrealistic if we compare to the present day Earth, except that Earth's estimated current CMB temperature is close to the mantle solidus. Yet our simulations can significantly deviate from present-day Earth's conditions. In the early stages of the simulations particularly, it is not rare to get melting in the lower mantle, and this may be realistic for the early Earth as people are now studying a long-lived basal magma ocean for example. Moreover, when the stagnant lid regime is reached for a long time, the mantle can be strongly insulated from the surface and sometimes warms up substantially. Since our simulations follow a very self-consistent design (nothing forces the evolution of the internal temperature), we have little control on what happens in the models, which explains why we chose to report observations in the present paper instead of attempting scaling laws of internal quantities. Certainly one can also use different solidus temperatures and also add the influence of water (a complete different problem) but of course this is not the point of this paper focused on the grain size evolution problem.

(3) In the lower mantle it is incorrect to talk about basalt, you should use "basaltic composition".

Yes sorry about that. This is unfortunately a common mistake that we do in our team as the "composition" is either "basalt" or "harzburgite" in the code. So we end up writing this in article. We have correct it.

(4) It 's impossible to see that "basalt is pushed aside". You need to have a figure with a zoom on the region of interest.

We add a comment on this in the text. Unfortunately we cannot load this figure even more.

Line 261: I do not understand what do you mean by "the newly formed parts of the pile". Since the pile does not entrain, but it is merely displaced,how do you generate "newly formed parts of the pile" ??
We have removed the density field plot from our paper since it does not provide significant interesting information as we do not focus our paper on the interaction of pile material with downgoing eclogitic material. Anyhow, due to the definition of "pile" parts of downgoing eclogite can become pile. Even though we do not observe large entrainment, it can be that small parts get entrained.

Line 268: Provide the solidus used to calculate melting at the CMB.
We now give the solidus in the text.

Line 270: It is wired: first we read that "basaltic material " melts up to 50%, and then we read "once the basaltic material has warmed up". How is this possible? Your statements are neither quantified nor justified. Show a P-T diagram with real temperatures and the used solidus for each composition, and then the reader will understand.
Yes, apologies, our observation was just wrong. In fact the material that melts was present before, close to solidus temperature and was decompressed by the return flow of the downwelling. Indeed the downwelling is cold and therefore is not melting. We have simplified the text as this was not really helping to make our point in the manuscript. This was indeed very confusing.

Line 293, line 295, and Figure 4: I do not understand why the pile density varies.
We have removed the text and the figure about density. This was not really helping to make any important point. To answer you: the density was varying because of both temperature and pressure changes. This happens because we do compressible convection. Plotting density was a little misleading because these adiabatic density

changes do not drive convection.

Line 301 and Figure 4: The modeled surface velocities can be higher than 10e3 cm/yr and up to 10e4 cm/yr, these values are huge (10-100 m/yr!!) and deserve a comment. Only saying "a lot of cold lithosphere simultaneously moves down" is insufficient. You need to quantify subducted volumes and you need to convince the reader that surface velocities at 10m/yr are not an artifact of the numerical simulation. We comment in the text. Yes these velocities are large but the load is much larger than present-day Earth's load. A 300km lithosphere destabilising as one plate would generate very large stresses. With a non-Newtonian (stress-dependent) rheology, such velocities make sense.

Line 310: I do not understand why the density of the pile changes because of "relocation" of pile material". Density variations caused by pressure variations are not an intrinsic density change, they are just an effect of compression/decompression. Indeed. Yes, this is also what we answered above. We have removed this confusing observation.

Lines 439 to 443: Rewrite.
It is rewritten.

Line 470: Provide reference of articles suggesting that piles "spatially determine subduction zones".
We have edited the paragraph to:
Our thermo-chemical piles are also not surrounded by plume generation zones (PGZ), as suggested by Burke et al. (2008), but plumes rise directly from the piles as well as from their margins. They, as others (Torsvik et al. (2006), Torsvik et al. (2010)), conclude that LLVPs (in geodynamics referred to as thermo-chemical piles) have been stable in time because the downward projection of Large Igneous Province

(LIP) sites can be linked to the margins of LLSVPs after rotating them back to their original eruption sites. LIPs in the 200 and 500 Myr age range let them conclude that LLSVPs have been occupying the same location for the same duration. Stable piles can only be confirmed with our models in case of absence of strong downwellings (subduction zones), hence for the last 200 to 500 Myr because we observe that downwellings govern the piles' spatial distribution. If there are no strong downwelling events disturbing the location of the piles, we can observe piles stable for at least 300 Myr. However, without dominant downwellings, we do not see plate tectonic-like behaviour in our simulations, implying that we either observe stable piles or plate tectonic-like behaviour, but not both simultaneously. Even without a plate tectonic-like convection regime in our models, it is difficult to draw conclusions about the actual stability and spatial distribution of LLSVPs. Problematic is that we neither employ realistic plate velocities, nor use three-dimensional models.

Line 509: Why is pile density self-regulating??
We have removed this.

Final comment: once you have reviewed the manuscript I suggest to rewrite parts of the abstract in a more concise, punchy, way.
We have rewritten parts of it.

---

## Editor Comment (EC1) · Julien Aubert (Editor) · 6 Feb 2020

I have now read through the replies to referee comments and the revised manuscript. The authors have done their best to answer most of the referee comments, but at the same time they do themselves acknowledge that several intrinsic problems of readability, conciseness and focus were not completely solved. One of these difficulties is in particular tied to the fact that there is another related manuscript in preparation, which raised difficulties to find the focus of the present manuscript.

I do however believe that the revision efforts are significant and the paper is a worthy addition to the topic and would consider it suitable for publication in SE as it now stands.

[Figure]

For future submissions I would encourage the authors to be more specific in their replies to referees, in particular by systematically mentioning the precise locations where changes are introduced in the text. Statement such as 'we have rewritten part of this section' are too vague and do not facilitate the evaluation of the improvements.

Please now submit the final manuscript so that I can proceed to acceptance.

Julien Aubert Topical Editor

―――――――――――――――――

---

## Author Response (AR2)

**Reply to reviewer #1 (Bradford Foley):**

**Major comment:**

The "self-regulating" nature of grain size in the piles is a major focus of the paper, and a regression for how the input parameters influence the resulting pile properties is used in part to demonstrate this self-regulating behavior. But I don't think the regression really contributes much to the results as presented. The models track the thermal evolution of the Earth, so the mantle and piles are thermally evolving over time. There are also changes in convection regime during the course of the models. Taking a time average over the whole model run time then really doesn't constrain the governing physics or necessarily even describe the system well. We don't have oscillations around a mean, but long-term temporal trends, so a simple average is obscuring these dynamic time evolving behaviors.

Moreover, average grain size in the piles in particular is pointed out as being insensitive to the varied input parameters in the regression shown in table 5. But the input parameters varied are not likely to cause grain size in the piles to vary. The parameters that control grain size are the grain growth activation energy and pre-exponential constant (not varied here), and the partitioning factor, f_G. f_top is varied, but not f_bot. The grain size in the piles will be much more sensitive to f_bot than f_top. None of the parameters varied and used in the regression are likely to affect grain size in the piles, so the finding that grain size is insensitive to these parameters does not illuminate the physics controlling how pile grain size evolves.

A better explanation of the long-term trends in pile grain size is still needed. It must be related to the temporal trends in deformational work rate and temperature, as these control the grain reduction and growth terms.

*Thanks for the comments.*
In order to answer this in detail, we wrote a new code to investigate the self-regulation of the viscosity in a semi-analytical manner. We add section 4.5 which illustrates in a different way what we mean by self-regulation and shows clearly the impact of fbot and the other parameters on the evolution of the LLSVPs in the Earth.
We leave the table with averages/regressions as we only use it as a 0-th order estimation of effective quantities in our parameter space. Indeed since several convection regimes occur, we did not attempt a more precise estimation. We chose to separately detail the characteristics of each convection regime instead.
The new section that we add is at the end of the discussion because we do not want to change the focus of the paper and make it more of an analytical study. But this addition definitely makes the message of our article more robust, thanks very much for the comment!

**Minor comments:**

Lines 9-10: "grain size dominates the viscosity development" and "viscosity can be dominated by temperature." This is unclear phrasing and should be re-worded. Also, Figure 6 shows both temperature and grain size controlling how viscosity in the piles evolves.
*We reformulated to:*
We further find, that although the average viscosity of the detected piles is buffered by both grain size and temperature, the viscosity is influenced predominantly by grain size. In the ambient mantle, however, depending on the convection regime, viscosity can also be predominantly controlled by temperature.

the viscosity is influenced predominantly by grain size
Line 209: "receive" is not the best word here; "determine" or "calculate" would work better
*We reformulated to:*
Then, we take the inverted sum of dislocation and diffusion creep viscosities to determine the total viscosity.

Line 223: Mulyukova & Bercovici 2017 also showed how grain growth activation energy changes the inferred temperature dependence of f_G
*We added the reference:*
Rozel et al., (2011) and Mulyukova and Bercovici, (2017) showed that f_G seems only to be temperature-dependent.

Pile detection: It's not clear why pile material must also be above the given temperature criteria. The example in figure 1 of "lost pile material" still looks attached to the pile. So why is it not counted?
*We added some additional explanation:*
We use this additional temperature constraint for the pile detection to prevent having piles that reach up too high from the CMB. Using these two constraints we start at the CMB with the detection. We check each cell-column for the criteria, moving upwards from the CMB. As soon as one of the criteria is not fulfilled, the pile top is reached. During this process, some pile material gets lost because it "overhangs" and is not attached to a continuous column starting at the CMB (figure 1).

Line 281: piles are strong due to "non-linearity of non-Newtonian fluids" but the pile rheology is typically diffusion creep rather than dislocation creep.
*We reformulated to:*
Panels a in figure 3 show that piles distribute around the large downwellings but and are not a strong layer which can prevent the downwelling material from reaching the CMB. Piles appear to be strong as long as no force acts on them, which can be attributed to the non-linearity of non-Newtonian fluids. However, piles nevertheless mainly deform in diffusion creep (figure 2e), even during their "strong" phase.

Line 306: Units of deformation work rate should be Pa/s instead of Pa*s
*We changed this.*

Line 333: "absent" should be "absence"
*Changed.*

Line 336: This statement is vague. Which properties and how similar are they?
A figure showing actual numbers would be a more clear way to make this
point. Though this statement also brings up the issue I raised above, about
whether a time average over a model with long-term temporal trends really
says much about the underlying physics governing the model results.
*We reformulated to:*
We observe that at the end of the simulations average properties of grain size,
viscosity, rheology, stress and strain rate are very similar to values in the first
billion year of the simulation despite the convection history.

Section 3.3: It would be much easier to see the temporal trends with plots
showing average mantle temperature (or potential temperature) and other
properties evolving over time, rather than trying to tease it out by looking at
1-D profiles plotted at different times
We decided to plot 1-D profiles because it provides a very quick and easy-to-
see overview on the differences throughout the whole Earth. We were not only
interested in the temporal evolution but also in the spatial evolution and
differences throughout the mantle.

Line 405: "succeed" should be "succeeded"
*Changed.*

Line 406: "constant stress" here is overly vague, since constant could refer to
constant in space, or constant stress over time. Wording needs to be clarified.
*We reformulated to:*
The rheology also shows, by a slightly higher component of diffusion creep
than recorded for the previously described time step, that grains are smaller
than at 2.5 Ga implying that the work rate has reduced compared to before.

Line 523-525: This discussion of viscosity in high melt fraction regions comes
out of nowhere; viscosity of melt fraction regions hasn't been discussed at all
up to now, so there's context missing for why this discussion is here. Where in
the model are the surprisingly high viscosities this sentence refers to?
*We moved this explanation to the beginning of the result section and removed*
*it from the conclusions. You are right, it was not the right place for this.*
*It now reads lines 271-275:*

[revised manuscript text omitted]